# DLO-Lab: Benchmarking Deformable Linear Object Manipulations with Differentiable Physics

**Junyi Cao** [1]  **Yian Wang** [1]  **Ziyan Xiong** [1]  **Chunru Lin** [1]  **Zhehuan Chen** [1]  **Chuang Gan** [1]

## Abstract

We address the challenge of enabling robots to manipulate deformable linear objects (DLOs), such as ropes, cables, and rubber bands. Prior work has primarily focused on narrow, task-specific problems, often relying on real-world demonstrations or handcrafted heuristics. Such approaches, however, struggle to scale to the wide variety of materials and tasks encountered in practice, and collecting sufficiently diverse real-world data is often impractical. Additionally, existing simulation environments offer limited support for the broad spectrum of material behaviors necessary for generalizable DLO manipulation. To overcome these limitations, we introduce a differentiable simulator explicitly designed for versatile DLO manipulation. Our simulator models a wide range of material properties—including (in)extensibility, elasticity, bending plasticity, and complex interactions with other objects—providing a robust foundation for learning and evaluating manipulation skills. Building on this simulator, we propose a benchmark suite of representative tasks that highlight the unique challenges of DLO manipulation. The successful execution of these tasks is often hindered by the topological complexity and grasp sensitivity inherent to DLOs. Therefore, we introduce a specialized DLO agent that explicitly manages these challenges by proposing strategic grasping points and decomposing long-horizon tasks to maximize control authority. Finally, we evaluate various policy-learning algorithms using our framework, alongside sim-to-real transfer experiments, demonstrating our platform's potential to advance DLO manipulation. Project page: https://dlo-lab-26.github.io/.

[1]University of Massachusetts Amherst. Correspondence to: Junyi Cao <junyicao@umass.edu>.

*Proceedings of the $43^{rd}$ International Conference on Machine Learning*, Seoul, South Korea. PMLR 306, 2026. Copyright 2026 by the author(s).

## 1. Introduction

Deformable object manipulation has long been a challenging topic in the field of robotic manipulation (Sanchez et al., 2018; Yin et al., 2021; Zhu et al., 2022; Hoang et al., 2025; Zhao et al., 2025; Wang et al., 2025; Huang et al., 2026; Li et al., 2026). Among the diverse forms of common deformable materials, deformable linear objects (DLOs), such as wires and ropes in Figure 1(a), exhibit unique features and properties. Compared to volumetric materials, a DLO can be represented by a relatively small set of vertices and degrees of freedom due to its linear co-dimensional nature. However, this apparent simplicity does not make manipulation any easier. Manipulating deformable linear objects (DLOs) presents several unique challenges: (1) high precision is essential due to the co-dimensional nature of these objects, (2) DLOs have complex dynamics and varied topologies that are challenging to model, and (3) perception algorithms often find it difficult to accurately detect the state of DLOs, especially when self-crossings happen.

Prior work has addressed DLO manipulation in several directions. (1) Approaches focusing on perception, such as cable-tracing tasks (Caporali et al., 2022; Kicki et al., 2023; Shivakumar et al., 2022; Viswanath et al., 2023; Luo & Demiris, 2025; Dinkel et al., 2025), which require extensive data collection and rely heavily on synthetic datasets for training. (2) Approaches focusing on manipulation, such as cable untangling (Viswanath et al., 2021; Sundaresan et al., 2021) and shape control (Yu et al., 2022a;b; Caporali et al., 2024; Gu et al., 2025), which generally target specific tasks and are difficult to generalize to other DLO-related problems. From this perspective, we argue that a simulation and benchmark platform for DLO manipulation is necessary—both for generating perception training data and for facilitating policy learning for manipulation.

Several prior works have proposed simulations for modeling DLO dynamics (Lin et al., 2021; Naughton et al., 2021; Li et al., 2021; Tong et al., 2023; Chen et al., 2024; Jiang et al., 2025). However, as shown in Table 1, each of them only supports a subset of the desired features. Among these features, coupling and differentiability are the two most important properties to enable flexible manipulation and efficient policy optimization. On one hand, coupling effectively models

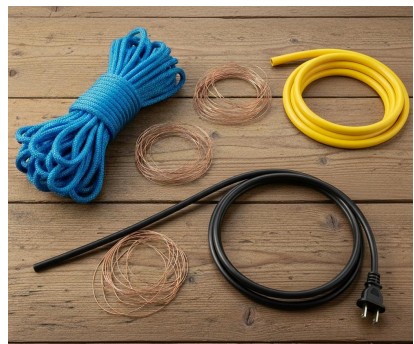 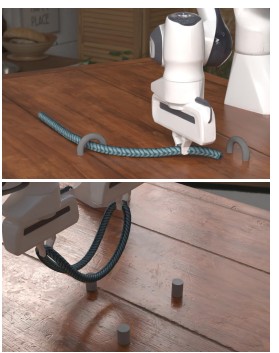 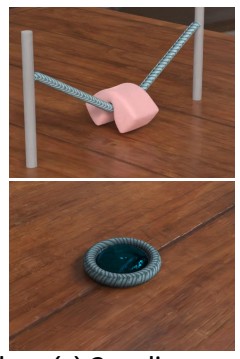 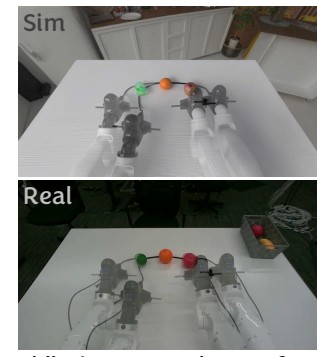

(a) Common DLOs in daily life    (b) Manipulation benchmark    (c) Coupling    (d) Sim-to-Real Transfer

*Figure 1.* **DLO-Lab.** (a) We encounter various deformable linear objects (DLOs) in our daily life, such as cables, ropes, and wires. (b) To facilitate versatile robotic skill learning for DLOs with diverse material properties, we introduce *DLO-Lab*, a differentiable simulation environment for DLOs with a set of benchmark tasks. (c) Our simulator effectively supports coupling with other materials, enabling the interaction between DLOs and other objects. (d) We also conduct real-world experiments to deploy a trained policy.

the physical interactions between DLOs, robot manipulators, and other objects in the scene. This physical fidelity is the foundation for simulating a wide range of complex, contact-rich manipulation tasks. On the other hand, differentiability is crucial for efficient policy optimization. It provides gradient information of the system's dynamics, which quantifies how changes in the robot's actions influence the final state of the DLO. However, none of the previous studies address *both* coupling and differentiability, and they also overlook other common properties found in DLOs, such as bending plasticity and loop topology. Therefore, we identify the need for a simulation environment that integrates diverse material couplers and is tailored for robotic skill acquisition in various DLO manipulation tasks.

To this end, we propose a differentiable simulation engine, DLO-Lab, featuring diverse material behaviors of DLOs and their interactions with other entities. Building upon the Genesis platform (Authors, 2024), we implement a customized solver for DLOs and their coupling with various materials, including rigid bodies, elastic and plastic objects, fluids, and beyond. Most operators in our solver are naturally differentiable by leveraging Taichi (Hu et al., 2019; 2020) as the programming language. Specifically, our DLO solver primarily follows the Discrete Elastic Rods (DER) theory (Bergou et al., 2008; 2010), enhanced with bending plasticity, support for loop topologies, and heterogeneous material composition. In addition, we implement frictional contact between DLOs and other materials and ensure that the differentiability requirement is satisfied during contact. As a result, our simulation is able to simulate the coupling between DLOs with other materials supported by Genesis, such as fluids and elastic objects, as shown in Figure 1(c).

Based on this simulation environment, we design a set of benchmark tasks that highlight the properties and capabilities of DLOs. Many of these tasks mimic practical applications like cable routing, wire forming, and unknotting,

which are inherently long-horizon and constrained by topology. In such scenarios, the success of a task also depends on identifying suitable grasping points and implementing strategic re-grasping, making simple end-to-end policy learning impractical. To tackle these challenges, we propose a specialized DLO agent that offers structural guidance through two main capabilities: (1) grasp proposal, which identifies optimal interaction points to maximize control authority, and (2) task decomposition, which breaks down complex objectives into manageable subtasks. We evaluate various algorithms on the benchmark tasks, including model-free reinforcement learning (MFRL) methods (Schulman et al., 2017; Haarnoja et al., 2018), first-order model-based reinforcement learning (FO-MBRL) methods (Xing et al., 2025; Xu et al., 2022), and sample- or gradient-based trajectory optimization (Hansen & Ostermeier, 2001).

Our experiments suggest that gradient-based methods achieve the highest sample efficiency when informative gradients are available, demonstrating the clear advantage of differentiable simulation. However, for tasks characterized by discontinuous contact dynamics or sparse rewards, sampling-based trajectory optimization proves more robust. We further observe that learning generalizable closed-loop policies remains significantly more sample-intensive than optimizing open-loop trajectories. Finally, real-world validation, *e.g.*, Figure 1(d), confirms that policies optimized in our simulator can be effectively transferred to physical hardware, successfully bridging the sim-to-real gap.

We summarize our main contributions as follows:

- We propose a differentiable simulation engine for DLOs and their interactions with different materials. Our simulation environment supports various material configurations and provides a standardized interface for both RL- and gradient-based policy learning algorithms.

- We introduce DLO-Lab, a comprehensive benchmark for

| Simulators | Solver | Elastic Potentials | Bending Plasticity | Loop Topology | Coupling | Differtiability |
|---|---|:---:|:---:|:---:|:---:|:---:|
| Bi-LSTM | NN | | | | | ✓ |
| GNN | NN | | | | | ✓ |
| XPBD | PBD | | | | | ✓ |
| SoftGym | PBD | | | | ✓ | |
| Elastica | Cosserat Rods | ✓ | | | ✓ | |
| C-IPC | DER | ✓ | | | ✓ | |
| IMC | DER | ✓ | | | | |
| DaXBench | MPM | ✓ | | | | ✓ |
| DEFORM | NN | ✓ | | | | ✓ |
| PhysTwin | Spring-Mass | ✓ | | | | ✓ |
| Ours | Customized | ✓ | ✓ | ✓ | ✓ | ✓ |

*Table 1.* **Comparison with existing simulators for DLOs.** "NN" refers to neural networks. "DER" refers to the Discrete Elastic Rods (Bergou et al., 2008; 2010) theory. "Coupling" refers to interaction between DLOs and other materials (*e.g.*, rigid or soft bodies).

robotic manipulation with DLOs. This benchmark aims to facilitate the exploration of versatile manipulation skills across a diverse set of DLOs. Based on DLO-Lab, we analyze the performance of sampling- and gradient-based policy learning methods and the challenges they encounter.

• We design a specialized agent for DLO manipulation tasks. It facilitates long-horizon manipulation by automatically decomposing complex tasks into executable sub-stages and proposing optimal grasping points to maximize control authority over the DLOs.

## 2. Related Work

**Differentiable Simulation** refers to computational frameworks where physical dynamics are fully differentiable with respect to system and control parameters. This paradigm has gained significant popularity in robotics and computer graphics, allowing for efficient, gradient-based optimization of policies (Mora et al., 2021; Huang et al., 2021; Xu et al., 2022; Xian et al., 2023; Wang et al., 2024) and system parameters (Li et al., 2022a; Xue et al., 2023; Ma et al., 2023; Cao et al., 2024; Lin et al., 2025) by back-propagating directly through the simulation. In general, these frameworks can be divided into two main categories. *Learning-based approaches* use neural networks to approximate the forward dynamics (Sanchez-Gonzalez et al., 2020; Pfaff et al., 2021; Chen et al., 2024) without explicit formulations. These methods are naturally differentiable through the networks and are typically trained on ground-truth simulation data to predict state transitions. Despite the efficiency, they require substantial training data and often have limited generalizability. *Analytical models*, on the other hand, adhere to classic simulation methods, *e.g.*, finite element methods (FEM) (Sifakis & Barbic, 2012) and material point methods (MPM) (Jiang et al., 2016), and calculate exact gradients of the underlying physical models. Some previous works, like DiffTaichi (Hu et al., 2020) and Warp (Macklin, 2022), use automatic differentiation with explicit time-stepping

schemes. Others, such as DiffSim (Qiao et al., 2020) and DiffCloth (Li et al., 2022b), derive analytical gradients and employ iterative solvers for implicit time integration.

**Deformable Linear Object Simulation** Recent advances in simulating DLOs use various solvers, yet none fully address the comprehensive requirements for robotic manipulation. These requirements encompass physical fidelity, coupling with different materials (rigid or soft), and differentiability. As summarized in Table 1, existing methods often necessitate trade-offs among these capabilities. Concretely, learning-based surrogates, *e.g.*, Bi-LSTM (Yan et al., 2020), GNN (Wang et al., 2022), DEFORM (Chen et al., 2024), and Position-based Dynamics (PBD) approaches, *e.g.*, XPBD (Liu et al., 2023), SoftGym (Lin et al., 2021), provide advantages like increased speed and differentiability. However, they often lack physical rigor and struggle to accurately model elastic energies and capture bending plasticity, limiting their realism in tasks involving stiff or permanently deformable materials. On the other hand, methods such as Elastica (Naughton et al., 2021), C-IPC (Li et al., 2021), and IMC (Tong et al., 2023), which rely on physically grounded models, can effectively capture elastic behaviors but often encounter significant integration challenges. Notably, these standard implementations typically lack differentiability, rendering them unsuitable for gradient-based policy optimization. Lastly, while differentiable solvers using Material Point Methods (MPM) or Spring-Mass systems, *e.g.*, DaXBench (Chen et al., 2023), PhysTwin (Jiang et al., 2025), provide gradient information, they often face challenges with robust coupling or lack support for complex topological constraints. Since no single existing solver combines all above features, we developed a customized solver to address these gaps for high-fidelity DLO manipulation.

**Deformable Linear Object Manipulation** represents a significant and long-standing challenge in robotics, with critical applications in industrial assembly, surgical robotics,

and household tasks (Saha & Isto, 2006; Lee et al., 2021; Laezza & Karayiannidis, 2021; Yu et al., 2022a; Lv et al., 2022; Zhaole et al., 2024; Li et al., 2025). Existing literature broadly categorizes DLO manipulation into two paradigms (Laezza & Karayiannidis, 2021; Laezza et al., 2021; Monguzzi, 2023). The first, explicit shape control, aims to deform the DLO into a precise geometric configuration. Works in this area often employ neural networks to learn the state transition of DLO dynamics, aiming to achieve a designated shape (Yan et al., 2020; Wang et al., 2022; Chen et al., 2024). In contrast, implicit shape control focuses on fulfilling high-level task conditions where the DLO's exact shape is not the primary objective. Such tasks include tying knots (Saha & Isto, 2006), routing cables (Luo et al., 2024), and fixing cables into clips (Li et al., 2018). Although they have achieved success in real-world applications, they are typically task-specific and difficult to generalize. Our work addresses this limitation by introducing a comprehensive simulation environment that facilitates learning versatile DLO manipulation policies.

## 3. Simulation Environment

In this work, we introduce a differentiable simulation environment tailored for DLO manipulation. We implemented the simulator using the Taichi (Hu et al., 2019; 2020) programming language and integrated it into Genesis (Authors, 2024) to build a versatile benchmark for DLO manipulation. Our simulator models a broad range of DLO's behaviors and its interactions with other rigid or soft bodies.

### 3.1. DLO Modeling

Following the classic discrete elastic rod (DER) (Bergou et al., 2008; 2010) methods, DLO-Lab represents a deformable linear object $\mathbf{X}$ as a centerline specified by a set of $N_v$ vertices and a set of $N_e$ adapted, orthonormal frames attached to each edge:

$$
\begin{aligned}
\mathbf{x} &= \{\mathbf{x}_i | \mathbf{x}_i \in \mathbb{R}^3\} \\
\mathbf{d} &= \{(\mathbf{d}_1, \mathbf{d}_2, \mathbf{d}_3)^j | \mathbf{d}_1, \mathbf{d}_2, \mathbf{d}_3 \in SO(3)\},
\end{aligned}
\tag{1}
$$

where $0 \leq i \leq N_v$ and $0 \leq j \leq N_e$. The term *adapted* means that $\mathbf{d}_3^j$ lies along the edge given by the adjacent vertices, $\mathbf{e}^j = \mathbf{x}_{j+1} - \mathbf{x}_j$, and $\mathbf{d}_3^j \equiv \mathbf{e}^j / |\mathbf{e}^j|$. During forward simulation, the internal interaction of DLOs is modeled by the potential energy $U(\mathbf{X}^t)$, where $\mathbf{X}^t$ denotes the state variables (*i.e.*, vertex states $\mathbf{x}^t$ and frame states $\mathbf{d}^t$) of the DLO at time $t$. Concretely, the potential energy is composed of stretching energy $U_s(\mathbf{d}^t)$, bending energy $U_b(\mathbf{x}^t)$, and twisting energy $U_t(\mathbf{x}^t)$. We calculate the derivatives $\partial U / \partial \mathbf{X}^t$ and employ a standard symplectic Euler solver for time stepping. In addition, we model the bending plasticity by introducing a yield threshold $\sigma_y$ and a creep rate $r_c$, which are used to adjust the rest curvatures when the yield

constraint is violated. Furthermore, we model the frictional contact behavior between DLOs using position-based dynamics (Müller et al., 2007), which simplifies the process of solving complex and computationally expensive non-linear equations typically required by preventative methods like IPC (Li et al., 2020; 2021). See Appendix A.1 for more details about the representation of DLOs used in this work.

### 3.2. Coupling Implementation

**Rigid Body** We adopt the rigid solver from Genesis (Authors, 2024) and implement a two-way coupling scheme between DLOs and rigid bodies. The rigid solver models each rigid geometry as a time-varying signed distance field (SDF). At each simulation step, we calculate the penetration depth $d$ of each rigid geometry for every sampled position $\mathbf{p}$ along the centerline of the DLO: $d(\mathbf{p}) = r(\mathbf{p}) - \text{SDF}(\mathbf{p})$, where $r$ is the radius. Rather than a hard binary contact, we use a soft-coupling approach where an exponential influence factor $f_i$, determined by a tunable softness parameter $\epsilon_s$, is calculated based on the penetration depth: $f_i = \min(\exp(d/\epsilon_s), 1)$. When $f_i$ exceeds a predefined threshold (0.1 as used in this work), we resolve the collision using an impulse-based response. The contact normal is derived from the gradient of the SDF, and we decompose the relative velocity between the sampled location of DLO and the contact point on the rigid body into normal and tangent components. This allows us to apply a standard restitution model to the normal component and a friction model to the tangential component. We calculate the change in the momentum at $\mathbf{p}$ and apply an equal and opposite reaction force back to the rigid body to ensure momentum conservation. Additionally, to stabilize the grasp with a rigid gripper, we identify the nearest vertices in contact with the manipulator and flag them as kinematic for the current time step. This prevents the DLO's internal position-based collision handling scheme from generating conflicting updates, ensuring that the DLO deforms compliantly around the gripper.

**Soft Body** To simulate the complex interactions between DLOs and other volumetric materials, such as fluids and elasto-plastic objects, we propose a two-way coupling scheme between our DLO solver and the Material Point Method (MPM) (Jiang et al., 2016; Hu et al., 2018) solver implemented in Genesis (Authors, 2024). We treat the interaction as a collision process mediated by the MPM Eulerian grid. Specifically, within the MPM grid operation loop, we detect collisions between active grid nodes and the DLO's geometry, represented by both its vertices and edges. When a grid node penetrates the DLO's collision volume, we calculate a repulsive impulse based on the relative velocity, surface normal, and material properties, such as restitution coefficients and local mass ratios. This impulse is applied symmetrically to ensure momentum conservation. It instantly updates the grid node's velocity to resolve the

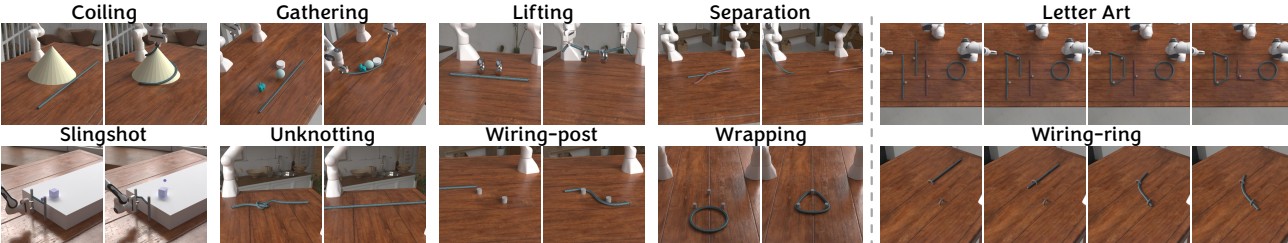

*Figure 2.* **Task illustration.** Our benchmark comprises 10 manipulation tasks: 8 fixed-horizon tasks shown on the left and 2 long-horizon tasks displayed on the right. The initial and desired goal states for each task are illustrated.

penetration of the soft body, while simultaneously applying an equal and opposite reaction force to the corresponding DLO vertices via atomic operations. This method facilitates stable, immediate bidirectional feedback between the discrete DLO elements and the continuum-based soft material within a single simulation step, enabling the simulation of multi-material phenomena, as illustrated in Figure 1(c).

### 3.3. Gradient Back-propagation

Our simulator, implemented with Taichi (Hu et al., 2019; 2020), benefits from its automatic differentiation (autodiff) to compute numerical gradients. However, a key challenge in achieving full differentiability in the simulation is enabling gradient flow throughout time. A naïve solution is to maintain a record of the simulation states at every timestep. Nevertheless, tasks in DLO-Lab often require thousands of simulation steps, rendering the naïve solution infeasible given limited GPU memory. To address this issue, we drew inspiration from FluidLab (Xian et al., 2023) and implemented gradient checkpointing to enable gradient backpropagation over the entire simulation trajectory. Specifically, to allow backpropagation through simulations of arbitrary length without being limited by GPU memory, we use gradient checkpointing. During the forward pass, we compute the simulation trajectory in segments. At the end of each segment, we cache a single state checkpoint in CPU memory and discard the intermediate GPU states. For the backward pass, we iterate through these checkpoints in reverse order. For each checkpoint, we rerun the forward simulation for the local segment to reconstruct the necessary computational graph, then perform the backward pass. This method balances the memory cost of storing the entire simulation history with the computational cost of recomputing small portions of the trajectory, keeping the memory requirement independent of the simulation horizon.

## 4. DLO-Lab Benchmark

Built upon the differentiable simulator, our DLO-Lab benchmark provides a versatile collection of DLO manipulation tasks equipped with differentiable reward functions. We also offer standardized APIs for building new manipulation environments. Our proposed tasks are illustrated in Figure 2.

### 4.1. Benchmark Representation

**Task Formulation** We address a series of manipulation tasks where an agent uses a set of robot arms equipped with parallel grippers to achieve specific goals. We formulate each task as a standard finite-horizon Markov Decision Process (MDP). Specifically, an MDP is comprised of a state space $\mathcal{S}$, an action space $\mathcal{A}$, a transition function $\mathcal{T} : \mathcal{S} \times \mathcal{A} \to \mathcal{S}$, and a reward function associated with each transition step $\mathcal{R} : \mathcal{S} \times \mathcal{A} \times \mathcal{S} \to \mathbb{R}$. We optimize the agent to derive a policy $\pi(a|s)$ that maximizes the expected sum of discounted rewards, $E_\pi \big[ \sum_{t=0}^{T} \gamma^t \mathcal{R}(s_t, a_t) \big]$, over a horizon $T$ with a discount factor $\gamma$.

**State Space** Let $N_v$ denote the total number of vertices of all DLOs in the scene, and let $N_m$ denote the total degrees of freedom (DoF) for all robot arms. The complete simulation state for the system is represented as: $\mathbf{S} = (\mathbf{x}, \dot{\mathbf{x}}, \mathbf{r}, \mathbf{M}, \dot{\mathbf{M}})$, where $\mathbf{x}, \dot{\mathbf{x}} \in \mathbb{R}^{N_v \times 3}$ gives the positions and velocities of all DLO vertices, $\mathbf{r} \in \mathbb{R}^{N_v \times 3}$ encodes the rest configurations, and $\mathbf{M}, \dot{\mathbf{M}} \in \mathbb{R}^{N_m}$ denote the joint positions and velocities of the robot arms.

**Observation** Unlike other deformable-material manipulation settings (Huang et al., 2021; Xian et al., 2023; Hoang et al., 2025) where downsampling is required to reduce the dimensionality of the observation space, in our case, it is computationally feasible to leverage the full state of the DLOs. Thus, the observation space for the DLOs consists of the stacked positions and velocities $(\mathbf{x}, \dot{\mathbf{x}})$, resulting in an $N_v \times 6$ representation. To capture the robot side of the interaction, we augment the observation with each manipulator's end-effector position $\mathbf{p}_i^{\text{ef}}$ and orientation $\mathbf{o}_i^{\text{ef}}$, along with the current joint configuration $\mathbf{M}$. When additional objects are presented, we also include the center-of-mass position and velocity of each object. This yields a complete observation vector that provides the agent with direct access to both the DLO dynamics and the manipulator kinematics, ensuring sufficient information for closed-loop control.

**Action Space** We implement end-effector pose control for the robot arms. At each step, the action specifies the gripper's target pose in Cartesian space. An inverse kinematics (IK) solver (Caron et al., 2025) is then used to determine the joint configurations required to reach these targets. This approach enables the policy to operate directly in task space,

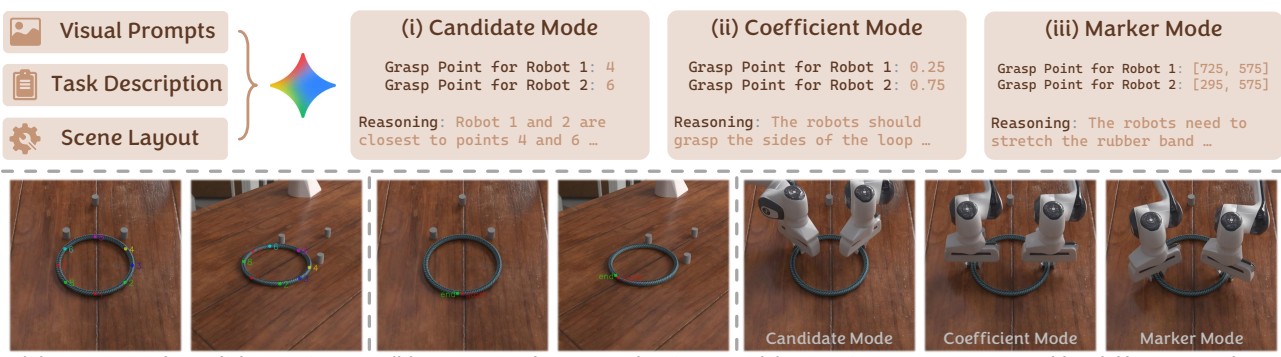

(a) Prompts of candidate points | (b) Prompts of start/end points | (c) Grasping points proposed by different modes

*Figure 3.* **DLO agent for grasping point proposal.** To decide the grasping points for robust DLO manipulation, we feed the DLO agent with task-related information and design three output modes, each with a different modality. We observe that *Candidate* mode yields the best reliability in our preliminary experiments (see Appendix C.3) and thus adopt it by default.

simplifying the robot's kinematics and ensuring the precise spatial alignment required for DLO manipulation.

## 4.2. Manipulation Tasks

**Coiling** The scene consists of a fixed cone and a rope. The agent needs to wind the rope around the cone's surface.

**Gathering** This task starts with several rigid and deformable bodies randomly placed on the ground. The objective is to use a rope to gather these objects together.

**Lifting** Given two "C"-shaped rings, the agent is required to manipulate a rope to lift the rings.

**Separation** In this task, the agent needs to use two robot arms to separate two ropes that are initially tangled together.

**Slingshot** In this scenario, a rigid ball and a rigid cube are placed on a table. In front of the ball, there is a slingshot made from a rope with high stretching stiffness. The agent should operate a robot arm and use the slingshot to launch the ball and hit the cube.

**Unknotting** This task requires the agent to untangle a rope that has an overhand knot.

**Wiring-post** The task involves manipulating a rope from a straight position into a specific "S"-shaped path that winds through two posts fixed to the table.

**Wrapping** The agent is asked to wrap a rubber band around three cylinders fixed to the table.

**Letter Art** The scene begins with three straight ropes, a fixed rubber band, and three posts fixed to the table. The agent should bend the three straight ropes to form the letters "D" and "L". When combined with the rubber band, the final arrangement should display "DLO".

**Wiring-ring** In this task, the agent needs to wire a rope through two rings fixed to the table.

Please see Appendix B.1-B.2 for additional details on task setups. We also relate the designed tasks to real-world manipulation skills in Appendix E.

## 4.3. DLO Agent

In DLO manipulation, the challenges of co-dimensional properties and underactuated dynamics can make it hard for end-to-end policies to effectively explore the action space, especially in long-horizon tasks with complex topologies. To tackle these issues, we present a specialized DLO agent integrating structural priors into the manipulation process.

**Grasp Proposal** The selection of grasping points is a critical determinant of success in DLO manipulation. In tasks such as *Unknotting*, selecting a suboptimal grasp can render the objective kinematically unsolvable. While manual specification of contact points is tedious and inefficient, random sampling is computationally prohibitive due to the vast search space. To address this, we leverage a Vision-Language Model (VLM) to propose grasping points guided by physical commonsense. As illustrated in Figure 3(d), we design three distinct prompting modes for this capability: (1) *Candidate* mode: The system uniformly samples points along the DLO, as in Figure 3(a), and queries the VLM to select the optimal candidate. (2) *Coefficient* mode: The VLM is provided with the DLO's endpoints, as in Figure 3(b), and prompted to output a scalar coefficient in $[0, 1]$, representing the normalized position of the grasp along the DLO's length. (3) *Marker* mode: The VLM is given visual markers for the DLO's endpoints, as in Figure 3(b), and is tasked with directly pinpointing the pixel coordinates of the suitable grasp on the rendered image. We evaluate these approaches in Appendix C.3 and find that *Candidate* mode consistently yields superior reliability and leads to better results. Thus, we adopt this mode for our subsequent experiments.

**Task Decomposition** Beyond the spatial challenge of grasping, the complexity of DLO manipulation is further compounded by the sequential dependencies inherent in long-horizon tasks. In these scenarios, a single continuous motion is often insufficient; the robot must strategically re-grasp the object and adapt to intermediate deformations. Consequently, standard end-to-end policy learning often struggles

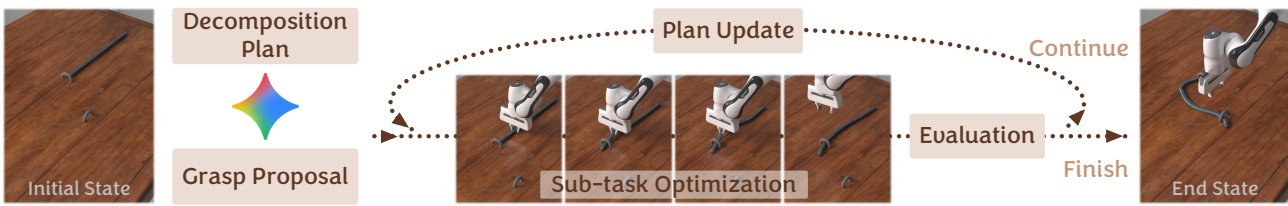

*Figure 4.* **DLO agent for task decomposition.** The DLO agent begins by proposing an initial task decomposition plan based on the need to change grasping points. We then perform iterative trajectory optimizations and provide the resulting trajectories to the agent for evaluation. It will continually update the sub-task plan and execute the next sub-task optimization until the task is deemed complete.

| Tasks | Coiling | Gathering | Lifting | Separation | Slingshot | Unknotting | Wiring-post | Wrapping |
|---|---|---|---|---|---|---|---|---|
| **PPO** | 9.40±0.07 | 39.76±0.12 | 247.38±0.20 | 114.31±13.21 | 6.90±0.00 | 3.29±0.00 | 62.17±1.97 | 131.08±7.20 |
| **SAC** | 8.28±0.74 | 40.76±0.53 | 250.29±2.44 | **134.71±0.93** | 7.23±0.23 | 2.95±0.03 | 62.07±0.73 | 161.85±1.59 |
| **SHAC** | 11.55±0.15 | 40.48±0.51 | 214.24±23.87 | 96.29±20.53 | 6.90±0.00 | 45.88±0.12 | 36.42±3.91 | 129.90±16.06 |
| **SAPO** | 11.57±0.07 | 40.29±0.33 | 204.54±18.70 | 105.27±14.87 | 6.90±0.00 | 46.30±0.49 | 36.13±2.83 | 144.36±8.18 |
| **GD** | 11.59 | 39.84 | 255.55 | 115.52 | 6.90 | 3.44 | 36.40 | 139.98 |
| **CMA-ES** | **11.73±0.02** | **47.84±0.35** | **335.59±14.21** | 84.86±0.40 | **11.07±0.43** | **57.21±1.51** | **64.31±0.56** | **162.68±0.86** |

*Table 2.* **DLO-Lab tasks tabular results.** We show the maximum episodic return within a fixed number of episodes and its standard deviation for the 8 fixed-horizon tasks over 3 random seeds. Since GD has no randomness in different trials, it has no standard deviation.

with the exploration problem, as fixed reward functions fail to effectively guide the agent through the extensive combinatorial space of possible actions. To mitigate this, we employ agentic task decomposition. As depicted in Figure 4, the process begins by prompting the VLM to generate an initial decomposition plan based on the task description. This plan explicitly defines the reward function and horizon length for each sub-task, along with the criteria for final success. Utilizing the grasp proposal, we then optimize each sub-task sequentially. A crucial step in this process is the closed-loop evaluation: after each execution, we render the optimized trajectory and ask the agent to assess whether the task has been completed. If the objective has not yet been achieved, the agent dynamically updates the plan for later phases based on the current state, ensuring the policy remains robust to execution errors. We evaluate the proposed agentic task decomposition on the two long-horizon tasks and present the results in Appendix C.4.

## 5. Experiment

Based on DLO-Lab tasks, we quantitatively evaluate three types of algorithms: (a) model-free reinforcement learning (MFRL) algorithms; (b) first-order model-based reinforcement learning (FO-MBRL) algorithms; and (c) trajectory optimization methods. We also conduct real-world experiments to verify the effectiveness of sim-to-real transfer.

### 5.1. Benchmarked Methods

We conduct a comprehensive performance evaluation of various methods. For MFRL algorithms, we evaluate Soft Actor-Critic (SAC) (Haarnoja et al., 2018) and Proximal

Policy Optimization (PPO) (Schulman et al., 2017). For FO-MBRL algorithms, we evaluate Short-Horizon Actor-Critic (SHAC) (Xu et al., 2022) and Soft Analytic Policy Optimization (SAPO) (Xing et al., 2025). For trajectory optimization methods, we consider the Covariance Matrix Adaptation Evolution Strategy (CMA-ES) (Hansen & Ostermeier, 2001) and gradient descent (GD).

### 5.2. Results and Analysis

**MFRL Algorithms** As shown in Table 2, MFRL methods generally underperform compared to trajectory optimization given the same sampling budget. This performance gap is structural: trajectory optimization solves a simpler open-loop control problem, whereas MFRL attempts to learn a generalized closed-loop policy. The latter carries a significantly higher burden, requiring extensive exploration and a "warm-up" phase before the policy network converges to meaningful behaviors. As a result, in the sample-constrained regime of complex DLO manipulation, MFRL is less sample-efficient than direct trajectory optimization.

**FO-MBRL Algorithms** Unlike standard MFRL, FO-MBRL methods, like SHAC and SAPO, leverage the simulator's differentiability to directly optimize policies using analytic gradients rather than relying solely on stochastic sampling. This provides a decisive advantage in contact-rich tasks where random exploration is insufficient. For instance, in the *Unknotting* task, which requires precise topological manipulation, SHAC and SAPO achieve an episodic return of ≈ 46, whereas PPO and SAC fail to make progress (returns ≈ 3). However, FO-MBRL still lags behind sample-based trajectory optimization, *i.e.*, CMA-ES, particularly in highly non-smooth or precisely coordinated tasks. This is

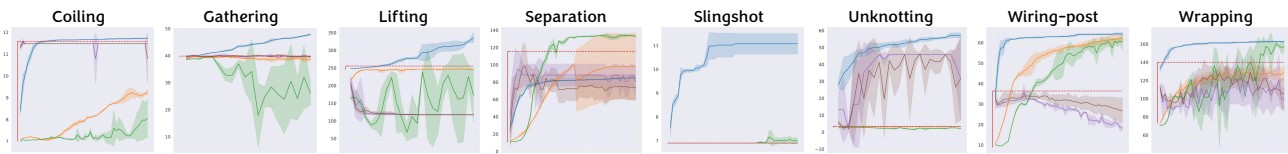

*Figure 5.* **DLO-Lab tasks training curves.** We report episodic return as a function of environment steps for the 8 fixed-horizon tasks. Color correspondence: **PPO**, **SAC**, **SHAC**, **SAPO**, **GD**, and **CMA-ES**. For trajectory optimization methods, we plot the prefix maximum.

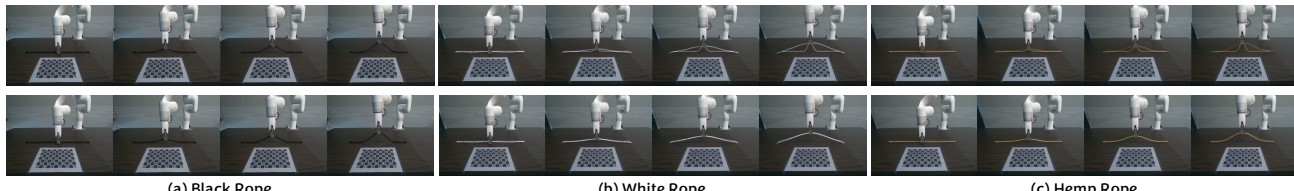

(a) Black Rope          (b) White Rope          (c) Hemp Rope

*Figure 6.* **System identification results for three different ropes.** We overlay the real-world captures and simulation renderings in this figure. *Top*: The simulation ropes with initial parameters. *Bottom*: The simulation ropes with optimized parameters following system identification. Our identified parameters effectively capture the physical properties of the real-world DLOs.

because learning a closed-loop policy presents extra challenges; gradients can be noisy or misleading, particularly during intermittent contact and topological transitions.

**Trajectory Optimization** The sample-based method, CMA-ES, achieves the strongest overall performance with reasonable sample efficiency, as shown in Table 2 and Figure 5. Its success can be attributed to two key factors. First, by bypassing the policy network, it avoids the optimization challenges associated with high-dimensional parameter spaces. Second, and more importantly, it does not depend on local gradient information, which can often be difficult or impossible to obtain for certain tasks (*e.g.*, *Lifting* and *Slingshot*). In such tasks, the reward signal depends on the interaction between DLOs and rigid objects. If the DLO is not yet touching the object, the gradient of the reward with respect to the robot's action is effectively zero. CMA-ES is immune to these issues and can use massive parallelization in simulation to effectively explore the landscape, making it a robust choice for complex, multi-stage manipulation.

While gradient-based trajectory optimization, GD, is theoretically efficient, our results reveal two major issues in practice. (1) Gradient inaccessibility: As noted above, in tasks requiring tool-use or indirect manipulation (*e.g.*, *Gathering*), useful gradients often do not exist until contact is established, making standard GD intractable without heuristic initialization. (2) Non-smooth landscapes: Even when gradients are available, the optimization landscape for DLOs is notoriously rugged due to discontinuous contact events and self-collisions. As a result, GD frequently gets trapped in local optima, as evidenced by low scores on tasks such as *Unknotting* (3.44) and *Wiring-post* (36.40). However, in tasks where the landscape is relatively smooth, and gradients are consistent—such as *Coiling* and *Separation*—GD demonstrates high efficacy. This validates the usefulness of differentiable simulation when the underlying physical

dynamics are suitable for first-order optimization.

### 5.3. Real-world Experiments

**System Identification** To bridge the gap between reality and simulation, we perform system identification to calibrate the physical parameters of the ropes used in our real-world experiments, including stretching and bending stiffness. We use our extrinsically calibrated camera setup to overlay simulation renderings directly onto real-world video footage for visual verification. Thanks to our differentiable simulator, our goal is simply to minimize the projection error between simulation and reality: we project the simulated rope onto the image plane to obtain a binary segmentation mask, and compute its pixel-wise difference with the binary mask extracted from the real-world captures. Because the simulator is differentiable, the gradients of this pixel-level projection error are back-propagated through the physics timesteps directly to the underlying material parameters. Figure 6 illustrates the validation of this process. The top row shows the simulation behavior using uncalibrated initialization parameters, with a noticeable discrepancy between the simulated and real-world ropes, particularly during high-deformation lifting phases. In contrast, the bottom row shows the results after parameter optimization, in which the simulation closely tracks the real-world ropes' deformation, exhibiting accurate geometric alignment and dynamic response. This strong correspondence confirms that our identified parameters effectively capture the physical properties of the real-world DLOs.

**Open-loop Policy Deployment** To validate the fidelity of our simulation, we conduct open-loop policy transfer on two tasks from DLO-Lab: (1) *Gathering*, performed using a Marvin M6 bimanual manipulator, and (2) *Wiring-post*, executed with an xArm. We employ a zero-shot strategy in which an open-loop policy is trained via sample-based tra-

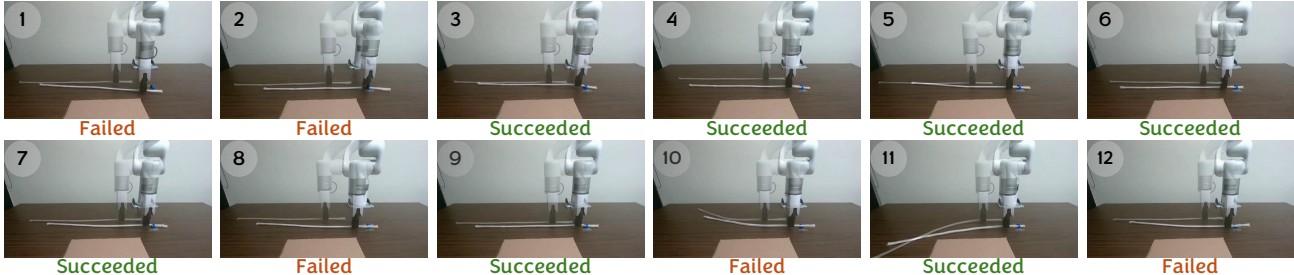

*Figure 7.* **Open-loop policy deployment results.**

| 1 | 2 | 3 | 4 | 5 | 6 |
| Failed | Failed | Succeeded | Succeeded | Succeeded | Succeeded |
| 7 | 8 | 9 | 10 | 11 | 12 |
| Succeeded | Failed | Succeeded | Failed | Succeeded | Failed |

*Figure 8.* **Closed-loop real-world policy deployment on the *Wiring-ring* task.** We show the initial (semi-transparent) and final (solid) states for each of 12 trials. **Succeeded** and **Failed** labels indicate the outcome of each trial.

jectory optimization in simulation and deployed directly on the physical hardware without additional fine-tuning. The process begins with system identification to calibrate the black rope's material parameters within the simulation (Figure 6(a)). During optimization, we incorporate the trajectory robustification technique proposed in SPIDER (Pan et al., 2025) to enhance policy reliability against disturbances. As shown in Figure 7, the trajectories generated in simulation are successfully executed on the physical setup, yielding outcomes comparable to the simulated results. These results confirm that our simulator captures sufficient physical realism to effectively bridge the sim-to-real gap.

**Closed-loop Policy Deployment** To demonstrate that DLO-Lab supports closed-loop sim-to-real transfer, we train a closed-loop policy (Schulman et al., 2017) on the *Wiring-ring* task and deploy it zero-shot on a physical xArm robot without additional fine-tuning. Here, *Wiring-ring* requires threading a rope through a ring fixed to the table, demanding precise, reactive manipulation. To bridge the perception gap, the policy is trained with a simulated sensor model that mirrors the real-world HSV tracker. At each step, clean 3D points are sampled from the rope centerline and ring surface, projected to 2D image coordinates, dilated using 4-connected pixel neighbors to simulate the objects' physical width, and then unprojected back to 3D with Gaussian depth noise. 384 points are randomly drawn from this combined noisy cloud to form the observation, so rope and ring pixels are treated jointly without separation. At deployment, the same 384-point cloud is obtained by applying HSV color thresholding to the RGB camera feed and back-projecting the detected pixels using known camera extrinsics. The policy therefore receives two inputs at each step: (1) the 384-point 3D point cloud described above; and (2) the robot's proprioception. As shown in Figure 8, the policy achieves

7 out of 12 successful completions (≈58%) with various initial rope positions. This result confirms that closed-loop policies trained in DLO-Lab can transfer to physical hardware and react responsively to the real-world rope state.

## 6. Conclusion

We introduced *DLO-Lab*, a differentiable simulation platform built on a customized physics solver unifying critical material behaviors, enabling efficient gradient-based policy learning for robotic DLO manipulation. We established a benchmark suite spanning real-world manipulation skills, from precise cable routing to topological unknotting. Addressing the long-horizon nature of these challenges, we developed a specialized DLO agent that employs hierarchical task decomposition and strategic grasp proposals to make complex planning feasible. Our extensive evaluation highlights the trade-offs between gradient-based and sampling-based optimization, offering key insights into policy learning for deformable objects. We validated the physical fidelity of our platform through zero-shot sim-to-real transfer in both open-loop and closed-loop settings: open-loop policies trained via trajectory optimization transfer directly to physical hardware for tasks such as gathering and wiring, while a closed-loop reactive policy achieves ≈58% success on the precision-demanding *Wiring-ring* task. Beyond single-task policies, DLO-Lab can serve as a scalable synthetic data engine: simulation demonstrations collected via trained policies in DLO-Lab enable fine-tuning VLA as a single multi-task policy across tasks, achieving results competitive with per-task RL baselines. We believe DLO-Lab provides a strong foundation for advancing learning-based DLO manipulation, with future opportunities in richer material models and large-scale sim-to-real generalization.

## Acknowledgements

This work was supported by NSF IIS-2441250, NSF IIS-2404386, MURI N000142412748, and NVIDIA's gift. We thank Yi-Ling Qiao and Minghao Guo for their insightful discussion during the preliminary stage of this work.

## Impact Statement

This paper presents work aimed at advancing the field of robotic manipulation, specifically for deformable linear objects. Our proposed simulation platform and agentic methods have potential downstream applications in industrial manufacturing (*e.g.*, cable harnessing) and healthcare (*e.g.*, surgical suturing). While this work contributes to the long-term goal of deploying robots in unstructured environments, we acknowledge that significant challenges remain. The broader societal implications of this research align with general considerations in robotics and automation, particularly regarding potential shifts in labor demand, and we do not foresee any unique ethical concerns or negative consequences beyond these established topics.

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

## A. Implementation Details

### A.1. DLO Representation

In this work, we mainly follow discrete elastic rod (Bergou et al., 2008; 2010) methods to model deformable linear objects (DLOs). Concretely, a DLO $\mathbf{X}$ is a centerline given by a set of $N_v$ vertices $\{\mathbf{x}_i\}_{i=1}^{N_v}$ and a set of $N_e$ orthonormal material frames $\{(\mathbf{d}_1, \mathbf{d}_2, \mathbf{d}_3)^j\}_{j=1}^{N_e}$ attached to each edge $\mathbf{e}^j = \mathbf{x}_{j+1} - \mathbf{x}_j$. Each vertex has its mass $m_i$ and radius $r_i$. Note that we use lower and upper indices to specify vertex- and edge-based properties, respectively. We assume $\mathbf{d}_3^j$ is adapted to the tangent $\mathbf{t}^j$ to the centerline, $i.e.$, $\mathbf{d}_3^j \equiv \mathbf{t}^j = \mathbf{e}^j/|\mathbf{e}^j|$. At the initial time $t = 0$, we predefine a reference frame $(\underline{\mathbf{d}}_1, \underline{\mathbf{d}}_2, \underline{\mathbf{d}}_3)$. The material frame can be obtained at a later time through a twist angle $\theta^j$ with respect to the reference frame. Thus, the total degrees of freedom (DoF) associated with the DLO, given by $N_v$ vertices and $N_e$ twist angles, is $3 \times N_v + N_e$.

To achieve elastic simulation for a DLO, we need to compute its elastic potential as defined by strain. Based on Kirchhoff's rod model (Dill, 1992; O'Reilly, 2017), strains can be separated into three categories: stretching, bending, and twisting.

- **Stretching** The stretching strain, or axial strain of an edge $\mathbf{e}^j$ is given by:

$$\epsilon^j = \frac{|\mathbf{e}^j|}{|\bar{\mathbf{e}}^j|} - 1, \tag{2}$$

where $|\bar{\mathbf{e}}^j|$ is the rest edge length. Denote $k_s^j = KA^j$ as the stretching stiffness, where $K$ is the stretching modulus and $A^j$ is the cross-sectional area, the stretching energy is then defined as:

$$U_s = \frac{1}{2} \sum_{j=1}^{N_e} k_s^j (\epsilon^j)^2 |\bar{\mathbf{e}}^j|. \tag{3}$$

- **Bending** The bending strain for an internal vertex $\mathbf{x}_i$ is defined by the curvature binormal $(\kappa\mathbf{b})_i$, which captures the misalignment between two adjacent edges:

$$(\kappa\mathbf{b})_i = \frac{2\mathbf{t}^{i-1} \times \mathbf{t}^i}{1 + \mathbf{t}^{i-1} \cdot \mathbf{t}^i}. \tag{4}$$

Note that $|(\kappa\mathbf{b})_i| = 2\tan(\phi_i/2)$, where $\phi_i$ is the bending angle between consecutive edges. The material curvature at an interior vertex is defined as

$$\boldsymbol{\kappa}_i = \frac{1}{2} \sum_{j=i-1}^{i} \left( (\kappa\mathbf{b})_i \cdot \mathbf{d}_2^j, (\kappa\mathbf{b})_i \cdot \mathbf{d}_1^j \right)^\top. \tag{5}$$

With these properties defined, we can formulate the bending energy as:

$$U_b = \frac{1}{2} \sum_{i=2}^{N_v-1} \frac{1}{\bar{l}_i} (\boldsymbol{\kappa}_i - \bar{\boldsymbol{\kappa}}_i)^\top B_i (\boldsymbol{\kappa}_i - \bar{\boldsymbol{\kappa}}_i), \tag{6}$$

where $\bar{l}_i = (|\bar{\mathbf{e}}^{i-1}| + |\bar{\mathbf{e}}^i|)/2$ is the Voronoi length for a vertex, $\bar{\boldsymbol{\kappa}}_i$ is the undeformed curvature,

$$B_i = \frac{EA_i}{4} \begin{pmatrix} r_i^2 & 0 \\ 0 & r_i^2 \end{pmatrix}, \tag{7}$$

is the bending stiffness, $E$ is the bending modulus.

- **Twisting** The twisting strain is defined as:

$$\tau_i = m_i - \bar{m}_i, \tag{8}$$

where $m_i = \theta^i - \theta^{i-1} + \underline{m}_i$, $\bar{m}_i$ is the undeformed twist, and $\underline{m}_i$ is the reference twist. Given the twisting stiffness $\beta_i = GA_i r_i^2/2$, where $G$ is the twisting modulus, the twisting energy writes as:

$$U_t = \frac{1}{2} \sum_{i=2}^{N_v-1} \beta_i \frac{(\tau_i)^2}{\bar{l}_i}. \tag{9}$$

**Inextensibility Constraint**    In addition, when inextensibility is desired, we can deactivate the stretching energy and use the following geometry projections to enforce the constraint. This approach directly modifies vertex positions to satisfy length constraints, offering greater stability than high-stiffness penalty forces. For each edge $\mathbf{e}^j$ connecting vertices $\mathbf{x}_j$ and $\mathbf{x}_{j+1}$, we first evaluate the constraint function $C(\mathbf{x}_j, \mathbf{x}_{j+1}) = |\mathbf{x}_{j+1} - \mathbf{x}_j| - |\bar{\mathbf{e}}^j|$, where $|\bar{\mathbf{e}}^j|$ is the rest edge length. A correction vector $\Delta\mathbf{x}$ is then calculated for each vertex, weighted by its inverse mass $w_i = 1/m_i$. The correction is distributed along the tangent vector $\mathbf{t}^j$, as shown in the following equations:

$$
\begin{aligned}
\lambda &= \frac{C(\mathbf{x}_j, \mathbf{x}_{j+1})}{w_j + w_{j+1}}, \\
\Delta\mathbf{x}_j &= \lambda w_j \cdot \mathbf{t}^j, \\
\Delta\mathbf{x}_{j+1} &= -\lambda w_{j+1} \cdot \mathbf{t}^j.
\end{aligned}
\tag{10}
$$

Finally, the vertex positions are updated via $\mathbf{x}_j = \mathbf{x}_j + \Delta\mathbf{x}_j$ and $\mathbf{x}_{j+1} = \mathbf{x}_{j+1} + \Delta\mathbf{x}_{j+1}$. This process is repeated for $N_C$ iterations to enforce the inextensibility of the DLO robustly.

**Bending Plasticity**    To simulate elastoplastic materials such as metal wires that can be permanently deformed, we keep track of the rest curvature $\bar{\boldsymbol{\kappa}}_i$ for each internal vertex $i$. Plastic deformation occurs when the magnitude of the elastic curvature $\boldsymbol{\kappa}_i^{el} = \boldsymbol{\kappa}_i - \bar{\boldsymbol{\kappa}}_i$ exceeds a predefined yield threshold, $\sigma_y$. If $|\boldsymbol{\kappa}_i^{el}| > \sigma_y$, the rest curvature is updated according to a creep model, which drives the rest curvature towards the current curvature at a rate proportional to the excess strain. The change in rest curvature, $\Delta\bar{\boldsymbol{\kappa}}_i$, for a given timestep is calculated as:

$$
\Delta\bar{\boldsymbol{\kappa}}_i = r_c \cdot \frac{|\boldsymbol{\kappa}_i^{el}| - \sigma_y}{|\boldsymbol{\kappa}_i^{el}|} \boldsymbol{\kappa}_i^{el},
\tag{11}
$$

where $r_c$ is the plastic creep rate. This update is then integrated over time, $\bar{\boldsymbol{\kappa}}_i = \bar{\boldsymbol{\kappa}}_i + \Delta\bar{\boldsymbol{\kappa}}_i$, allowing the rod to retain a new intrinsic shape after significant bending.

**Loop Topology**    To properly initialize the material frames for a DLO with a closed-loop topology, we must account for the geometric phase, or holonomy, that arises from parallel transport around a non-trivial curve. A naive sequential transport of the material frame from the first edge ($j = 1$) to the last ($j = N_e$) results in a discontinuity, as parallel transporting the final frame $\bar{\mathbf{d}}_1^{N_e}$ back across the closing edge to the tangent of the first edge, $\mathbf{t}^1$, will not align with the original frame $\bar{\mathbf{d}}_1^1$. To resolve this, we first compute the total holonomy angle, $\psi_H$, which represents this angular mismatch. We then distribute this error evenly across all edges. For each edge $j$, we compute a local correction angle $\phi^j = -\psi_H \cdot (j - 1)/N_e$. The initial, uncorrected material frame vectors, denoted as $\hat{\mathbf{d}}_1^j$ and $\hat{\mathbf{d}}_2^j$, are then rotated by $\phi^j$ to obtain the final, continuous reference frames for the closed loop:

$$
\begin{bmatrix} \bar{\mathbf{d}}_1^j \\ \bar{\mathbf{d}}_2^j \end{bmatrix} = \begin{bmatrix} \cos\phi^j & \sin\phi^j \\ -\sin\phi^j & \cos\phi^j \end{bmatrix} \begin{bmatrix} \hat{\mathbf{d}}_1^j \\ \hat{\mathbf{d}}_2^j \end{bmatrix}.
\tag{12}
$$

This procedure ensures that the material frame is consistent and continuous across the entire loop at its rest state.

**Self-collision and Friction**    To handle collisions and frictional contact among DLOs, we employ a hybrid PBD approach that combines geometric projections for non-penetration with velocity-level updates for friction. We model each segment as a continuous cylinder with radius $r_i$. For any pair of potentially colliding edges, defined by vertices $(\mathbf{x}_i, \mathbf{x}_{i+1})$ and $(\mathbf{x}_j, \mathbf{x}_{j+1})$, we first compute the closest points between the two line segments. These points, $\mathbf{c}_a$ and $\mathbf{c}_b$, are found using barycentric coordinates $t$ and $u \in [0, 1]$ such that $\mathbf{c}_a = (1-t)\mathbf{x}_i + t\mathbf{x}_{i+1}$ and $\mathbf{c}_b = (1-u)\mathbf{x}_j + u\mathbf{x}_{j+1}$. The non-penetration constraint is enforced if the penetration depth $d = (r_a + r_b) - |\mathbf{c}_a - \mathbf{c}_b|$ is positive. A correction vector is then applied to the four defining vertices, weighted by their respective inverse masses and barycentric coordinates. The correction for vertex $\mathbf{x}_i$, for example, is calculated as:

$$
\Delta\mathbf{x}_i = \frac{d \cdot w_i(1-t)}{w_i(1-t)^2 + w_{i+1}t^2 + w_j(1-u)^2 + w_{j+1}u^2} \cdot \mathbf{n},
\tag{13}
$$

where $\mathbf{n} = (\mathbf{c}_a - \mathbf{c}_b)/|\mathbf{c}_a - \mathbf{c}_b|$ is the contact normal. We also repeat the geometric projections for $N_C$ iterations. After the positional correction, friction is resolved at the velocity level. We implement a Coulomb friction model where the tangential velocity update $\Delta\mathbf{v}_t$ is proportional to the normal impulse magnitude, which is approximated from the penetration depth $d$. This velocity correction is then distributed to the four vertices, similarly weighted by their inverse masses and barycentric coordinates, to provide a stable and physically plausible contact response.

| Tasks | #Vertices | Stretching Modulus | Bending Modulus | Twisting Modulus | Inextensibility | Plasticity |
|---|---|---|---|---|---|---|
| Coiling | 60 | – | 1e3 | 1e3 | ✓ | – |
| Gathering | 45 | 1e5 | 1e3 | 1e3 | – | – |
| Lifting | 30 | 1e5 | 5e3 | 1e3 | – | – |
| Separation | 30 | – | 5e3 | 1e3 | ✓ | – |
| | 30 | – | 5e3 | 1e3 | ✓ | – |
| Slingshot | 12 | 8e5 | 1e5 | – | – | – |
| Unknotting | 50 | 1e5 | 1e4 | – | – | – |
| Wiring-post | 30 | – | 1e4 | 1e3 | ✓ | – |
| Wrapping | 50 | 1e5 | 1e4 | – | – | – |
| | 24 | 1e5 | 1e4 | – | – | ✓ |
| Letter Art | 30 | 1e5 | 1e4 | – | – | ✓ |
| | 25 | 1e5 | 2e3 | – | – | ✓ |
| Wiring-ring | 30 | 1e5 | 1e4 | – | – | – |

*Table 3.* **Parameter settings of DLOs used in our simulation benchmark tasks.**

## A.2. Rendering

We utilize the customized LuisaRender (Zheng et al., 2022) available in Genesis to create photorealistic images of our simulation results. The enhanced photorealism enables our simulation environment to generate perception training data for policy learning in DLO manipulation, paving the way for future applications.

## B. Environmental Setup

### B.1. Task Setup

In all our manipulation tasks, we set the frame time equal to 1e-3 seconds, and each frame is computed using 5 substeps. For the 8 fixed-horizon tasks, we set the number of frames to 2,000, except for the Slingshot task, where we set it to 800. We report the parameter settings for each simulation task in Table 3.

### B.2. Reward Design

Here, we outline the reward design used for our manipulation tasks.

**Coiling**    The reward is defined as the sum of the negative distances from each vertex of the rope to the center of the cone.

**Gathering**    The reward consists of three parts. (a) Object clustering: The negative sum of pairwise distances between the three target objects. (b) Rope-object contact: The negative minimum distance from the rope vertices to each object. (c) Penalty for naïve solution: $-\sum_k \sum_o \max(0, \epsilon - d(\mathbf{p}_k^{\mathrm{ef}}, \mathbf{p}_o))$, where $\epsilon = 0.1$ is the margin, $\mathbf{p}_k^{\mathrm{ef}}$ denotes the $k$-th end-effector position, and $\mathbf{p}_o$ denotes the center-of-mass position of the $o$-th target object. This term discourages the agent from directly using end-effectors to push the target objects together.

**Lifting**    We define the reward function as the sum of four components. (a) Elevation: The sum of the vertical positions of the two target rings. (b) Contact: The negative minimum distance between the rope vertices and the center of each ring. (c) Lifting configuration: A positive reward for positioning the engaging rope vertices slightly above the center of each ring, promoting a stable lifting configuration. (d) Penalty for naïve solution: $-\sum_k \sum_r \max(0, \epsilon - d(\mathbf{p}_k^{\mathrm{ef}}, \mathbf{p}_r))$, where $\epsilon = 0.08$ is the margin, $\mathbf{p}_k^{\mathrm{ef}}$ denotes the $k$-th end-effector position, and $\mathbf{p}_r$ denotes the center position of the $r$-th ring. This term enforces indirect manipulation via the rope.

**Separation**    We calculate the Chamfer distance between the two ropes and use it as the reward.

**Slingshot**    The reward is defined as the moving distance of the cube projected onto the $+y$-axis (the direction away from the slingshot) after the slingshot is released.

**Unknotting**    We compute a penalty term composed of two parts, and the reward is the negative of the penalty. (a)

Topological complexity: We calculate the discrete Average Crossing Number (ACN) by summing the signed crossing contributions of all non-adjacent edge pairs. (b) Self-repulsion: To prevent the rope from intersecting itself or forming tight, physically implausible clusters during manipulation, we add a heavy penalty for any non-adjacent segments that approach within twice the segment interval.

**Wiring-post**   We define the reward as the sum of three components. (a) Shape alignment: We compute the negative Chamfer distance between the rope vertices and the goal-state positions. (b) Midpoint anchoring: We encourage the midpoint of the rope to align with a designated center point on the $xy$-plane, with a 2cm tolerance. (c) Height penalty: A penalty is applied if any part of the rope exceeds the height of the posts.

**Wrapping**   The reward consists of two parts. (a) Topological winding: we compute the discrete Winding Number for the rope relative to each post by summing the signed angles between consecutive rope segments in the $xy$-plane. The reward is maximized when the magnitude of this winding number approaches 1.0 (*i.e.*, one full revolution) for every post. (b) Surface contact: To tighten the loop, we penalize the minimum distance between the rope and each post.

Note that we apply the exponential or translational transformation to the reward function when necessary to ensure it is non-negative. The loss used for gradient-based methods is typically defined by the negative rewards in those tasks.

For **Letter Art** and **Wiring-ring**, the reward for each sub-task is obtained from the DLO agent. Please see Appendix F for detailed prompts.

| actor lr | 5e-4 |
|---|---|
| actor mlp | [256, 128, 64] |
| critic lr | 5e-4 |
| critic mlp | [256, 128, 64] |
| grad clip | 0.5 |
| GAE $\lambda$ | 0.95 |
| entropy coeff. | 5e-3 |

*Table 4.* **PPO parameters.**

| actor lr | 5e-4 |
|---|---|
| actor mlp | [256, 128, 64] |
| critic lr | 5e-4 |
| critic mlp | [256, 256] |
| grad clip | 0.5 |
| replay buffer | 2e5 |
| target entropy | $-\dim(\mathcal{A})/2$ |

*Table 5.* **SAC parameters.**

| actor lr | 2e-3 |
|---|---|
| actor mlp | [256, 128, 64] |
| critic lr | 5e-4 |
| critic mlp | [256, 256] |
| grad clip | 0.5 |
| GAE $\lambda$ | 0.95 |
| short horizon | 20 |

*Table 6.* **SHAC parameters.**

| actor lr | 2e-3 |
|---|---|
| actor mlp | [256, 128, 64] |
| critic lr | 5e-4 |
| critic mlp | [256, 256] |
| entropy lr | 5e-3 |
| grad clip | 0.5 |
| GAE $\lambda$ | 0.95 |
| short horizon | 20 |

*Table 7.* **SAPO parameters.**

### B.3. Model-free Reinforcement Learning Setup

We use open-source implementations (D'Eramo et al., 2021) of PPO (Schulman et al., 2017) and SAC (Haarnoja et al., 2018) in our environments. We set the batch size to be $2,048$ for both PPO and SAC. Below, we provide other hyperparameters in Table 4-5. Note that $\mathcal{A}$ in Table 5 denotes the dimension of the action space.

### B.4. First-order Model-based Reinforcement Learning Setup

We use open-source implementations [1] of SHAC (Xu et al., 2022) and SAPO (Xing et al., 2025) in our environments. We generally follow the default settings from their original paper and list the hyperparameters in Table 6-7.

### B.5. Trajectory Optimization Setup

We use the open-source implementation (Hansen et al., 2019) of CMA-ES. We set the population size to 400 and the initial sigma value to 5e-2 (except for *Gathering*, *Lifting*, and *Letter Art*, which use 5e-3). For gradient descent, we use the Adam optimizer (Kingma, 2014) with an initial learning rate of 1e-3 (except for *Gathering*, *Lifting*, and *Unknotting*, where it is set to 1e-4), which decreases to 1e-6 using a cosine schedule. We adopt an early-stopping strategy for gradient descent, terminating the optimization after 10 episodes if the loss does not decrease, as it often gets stuck in local minima.

## C. Additional Results

### C.1. Training Curves based on Wall-Clock Time

In Figure 5, we present the training curves based on environment steps. Here, we plot these curves as a function of wall-clock time in seconds. The results are shown in Figure 9. It is evident that when informative gradients are available, the gradient-based trajectory optimization method (GD) can converge to a local minimum in just a few minutes. In contrast,

---

[1]https://github.com/etaoxing/mineral

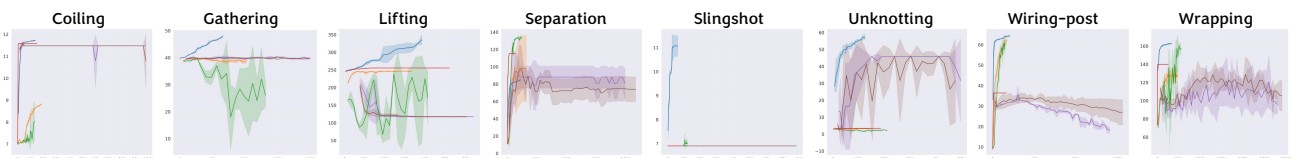

*Figure 9.* **DLO-Lab tasks training curves.** We report episodic return as a function of wall-clock time in seconds for the 8 fixed-horizon tasks in our benchmark. Color correspondence: **PPO**, **SAC**, **SHAC**, **SAPO**, **GD**, and **CMA-ES**. For trajectory optimization methods, we plot the prefix maximum.

| Tasks | Coiling | Gathering | Lifting | Separation | Slingshot | Unknotting | Wiring-post | Wrapping | Average |
|---|---|---|---|---|---|---|---|---|---|
| **PPO** | 67% | 0% | 0% | 100% | 0% | 0% | 67% | 0% | 29.3% |
| **SAC** | 0% | 0% | 0% | 100% | 33% | 0% | 0% | 0% | 16.6% |
| **SHAC** | 93% | 0% | 0% | 100% | 0% | 73% | 0% | 0% | 33.3% |
| **SAPO** | 100% | 0% | 0% | 100% | 0% | 80% | 0% | 0% | 35.0% |
| **GD** | 100% | 0% | 0% | 100% | 0% | 0% | 0% | 0% | 25.0% |
| **CMA-ES** | 100% | 100% | 87% | 100% | 93% | 93% | 93% | 27% | 86.6% |

*Table 8.* **DLO-Lab tasks success rates.**

| Tasks | Lifting | Unknotting | Wrapping |
|---|---|---|---|
| **Candidate Mode** | **335.59±14.21** | **57.21±1.51** | **162.68±0.86** |
| **Coefficient Mode** | 330.57±9.03 | 3.06±0.01 | 144.78±3.23 |
| **Marker Mode** | 330.57±9.03 | 3.06±0.01 | 136.39±1.14 |

*Table 9.* **Comparison of different modes of the DLO agent for grasp proposal.** We use CMA-ES with the grasping points suggested by different modes and report the maximum episodic return. For other tasks, the suggested grasping points are generally the same.

the sample-based trajectory optimization method typically requires one to two hours to produce a satisfactory trajectory, although the final result often outperforms GD. In reinforcement learning (RL), the process typically takes several hours to tens of hours to develop a closed-loop policy.

## C.2. Success Rates

To provide an intuitive understanding of the performance of different methods on our benchmark tasks, we assess their success rates in Table 8. Specifically, for RL algorithms, *i.e.*, PPO, SAC, SHAC, and SAPO, we evaluate 15 trajectories for each algorithm across each task. In the case of GD, which is deterministic, we assess the final optimized trajectory for each task. For CMA-ES, we select the top 15 trajectories based on the returns from the best checkpoint. We determine the success of a trajectory by checking whether the agent achieves its goal at the end of the rendered videos.

## C.3. Comparison on Grasp Proposal

We quantitatively evaluate the efficacy of the three grasp proposal modes, *i.e.*, *Candidate*, *Coefficient*, and *Marker*, in Table 9. To ensure fair comparison, the raw outputs from the *Coefficient* and *Marker* modes are post-processed by projecting the predicted locations onto the nearest DLO vertices to obtain valid 3D grasping coordinates. As evidenced by the results, *Candidate* mode consistently outperforms the other strategies, achieving the highest episodic returns across all tested tasks. The performance gap is most pronounced in the *Unknotting* task, where precision is paramount. We attribute this superiority to the formulation of the problem: *Candidate* mode constrains the VLM's decision space to a discrete set of physically valid points, effectively transforming the challenge from a continuous spatial estimation task, which is prone to hallucination and projection errors, into a more robust selection task. Conversely, the *Coefficient* and *Marker* modes often suffer from minor spatial inaccuracies that, after snapping to the nearest vertex, result in suboptimal grasping policies.

## C.4. Analysis of Task Decomposition

We evaluate our DLO agent for task decomposition in the *Letter Art* and *Wiring-ring* tasks with the sample-based trajectory optimization method, CMA-ES, in Figure 10. To validate the necessity of the dynamic plan update strategy, which closes

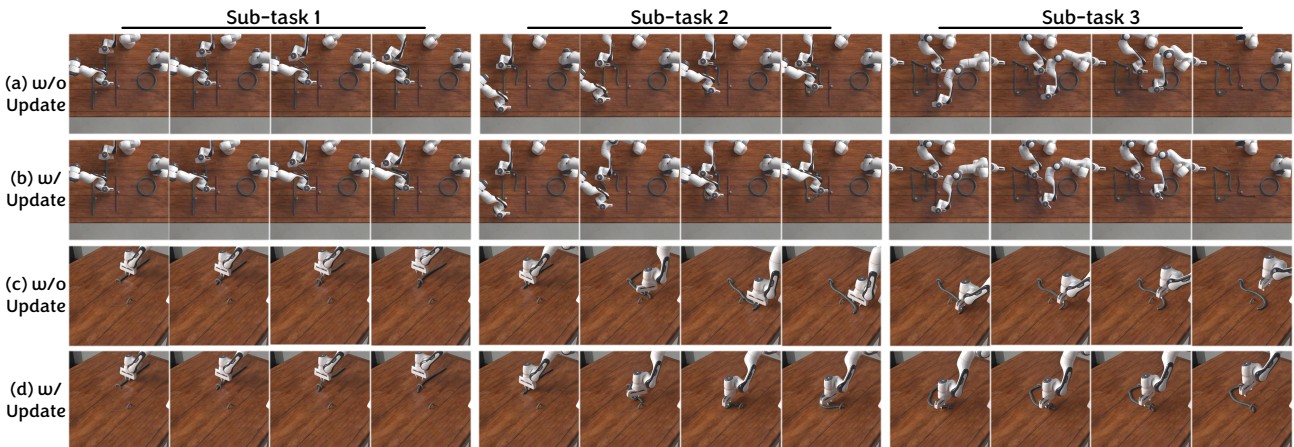

*Figure 10.* **Analysis on the plan update strategy of the DLO agent for task decomposition.** We analyze the plan update strategy of the DLO agent in the (a-d) *Letter Art* and (c-d) *Wiring-ring* tasks. Without the plan update, the task decomposition plan is created at the start and remains fixed throughout the task. When the plan is updated after completing each sub-task, the agent can adjust sub-task rewards and dynamically re-plan sub-tasks, helping achieve the overall goal more effectively.

the loop between high-level planning and low-level execution, we also compare the trajectories of the DLO agent with and without this feedback mechanism, *i.e.*, rows (a)&(c) vs. rows (b)&(d). The results show that for tasks with spatially distinct and independent sub-goals, such as *Letter Art* (a-b), the initial decomposition plan is sufficient and yields comparable performance. However, in sequential, precision-critical tasks like *Wiring-ring* (c-d), this static approach is inadequate. Concretely, in Figure 10(c), the agent successfully threads the first ring but struggles to transition to the second ring. To explain, this difficulty arises from the rigidity of the pre-computed reward, which does not account for the rope's deformed state after the first interaction. In contrast, the dynamic plan update allows the agent to re-assess the intermediate state and develop highly specific reward functions tailored to the immediate manipulation phase. As observed in our experiments, after completing sub-task 1, the agent dynamically refines the objective for the second ring by dividing it into distinct phases. First, it generates a "corridor" reward (sub-task 2) to ensure precise radial alignment and a stable approach. This is followed by a "tunneling" reward (sub-task 3) that specifically incentivizes axial pushing while penalizing buckling. This adaptive re-planning maintains a dense, informative local optimization landscape, effectively correcting execution drift in long-horizon tasks.

### C.5. Data Generation and VLA Fine-tuning

Beyond single-task optimization, DLO-Lab can serve as a high-fidelity data engine for training generalizable visuomotor policies. We demonstrate this by collecting simulation demonstrations and fine-tuning a Vision-Language-Action (VLA) model, then evaluating it zero-shot as a single multi-task policy across four tasks from our benchmark simultaneously.

**Data Collection** Using the trained policies in DLO-Lab, we generate 200 successful demonstrations per task across four tasks—*Coiling*, *Separation*, *Unknotting*, and *Wiring-post*—yielding 800 demonstrations in total. Each episode records rich multi-view visual observations comprising one front-facing camera and one to two wrist cameras depending on the task (see Figure 11), paired with the corresponding end-effector control actions in DLO-Lab's standardized controller format.

**Model and Training** We fine-tune SmolVLA (Shukor et al., 2025), an efficient open-source Vision-Language-Action model, on the combined 800-demonstration dataset with a batch size of 128 for 40K iterations. Crucially, a *single* SmolVLA policy is trained jointly across all four tasks, whereas the RL baselines (PPO, SAC, SHAC, SAPO) each require a dedicated policy trained from scratch for each task.

**Results** We evaluate SmolVLA on 15 trajectories per task and report success rates in Table 10. The single multi-task SmolVLA policy achieves an average success rate of 60.0%, competitive with per-task PPO (58.5%) and substantially higher than per-task SAC (25.0%). On *Separation* and *Unknotting*, SmolVLA matches the best RL methods, while *Wiring-post* remains challenging due to the high precision required for tip insertion. These results demonstrate that DLO-Lab can function as a scalable synthetic data engine to bootstrap generalizable manipulation policies.

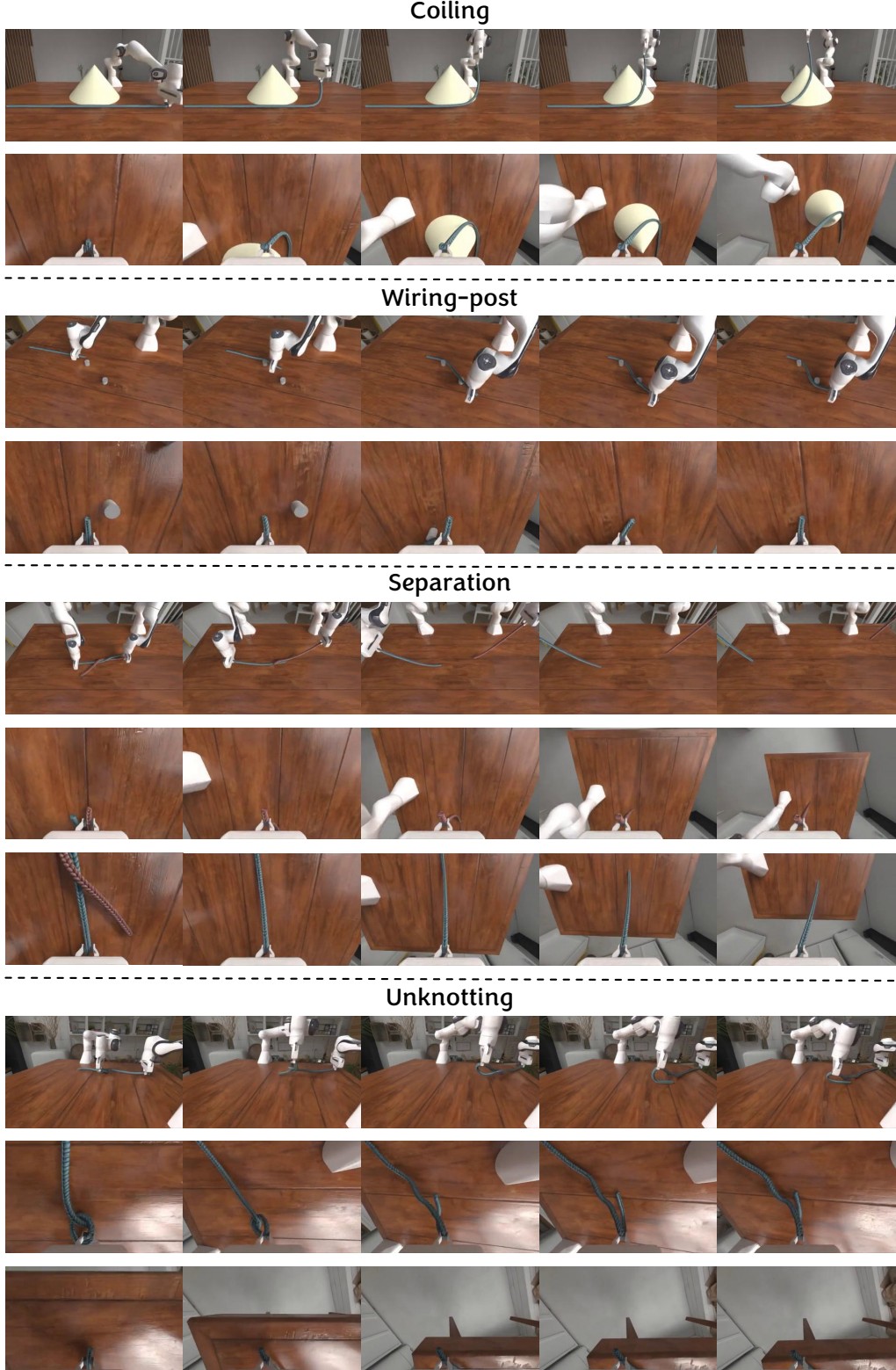

*Figure 11.* **Simulation demonstrations for VLA fine-tuning.** For each task, we display observations from the front camera (top row) and wrist camera(s) (subsequent rows).

| Method | Coiling | Separation | Unknotting | Wiring-post | Average |
|---|---|---|---|---|---|
| **PPO** | 67% | 100% | 0% | 67% | 58.5% |
| **SAC** | 0% | 100% | 0% | 0% | 25.0% |
| **SHAC** | 93% | 100% | 73% | 0% | 66.5% |
| **SAPO** | 100% | 100% | 80% | 0% | 70.0% |
| **SmolVLA (multi-task)** | 60% | 100% | 73% | 7% | 60.0% |

*Table 10.* **Success rates for VLA fine-tuning.** RL baselines are per-task single-task policies. SmolVLA is evaluated as a single policy trained jointly on all four tasks. Each method is evaluated on 15 trajectories per task.

| Tasks | #Envs | Coiling | Gathering | Lifting | Separation | Slingshot | Unknotting | Wiring-post | Wrapping |
|---|---|---|---|---|---|---|---|---|---|
| **Forward** | 1 | 90.85 | 32.31 | 50.03 | 87.20 | 90.08 | 80.70 | 90.82 | 83.71 |
| | 10 | 873.73 | 175.79 | 418.94 | 819.08 | 829.03 | 725.73 | 883.78 | 752.17 |
| | 50 | 4349.98 | 365.86 | 1736.94 | 3372.44 | 3759.83 | 3192.53 | 4125.65 | 3327.49 |
| | 100 | 8566.86 | 422.92 | 2625.83 | 4939.99 | 6120.69 | 5131.42 | 8531.82 | 5139.43 |
| **Forward &** | 1 | 32.49 | 8.24 | 12.10 | 30.29 | 13.73 | 28.68 | 31.13 | 31.17 |
| | 10 | 344.25 | 52.00 | 53.37 | 267.77 | 106.72 | 211.16 | 282.10 | 186.84 |
| **Backward** | 50 | 1465.67 | 184.28 | 153.30 | 1056.34 | 412.05 | 709.75 | 1382.38 | 670.37 |
| | 100 | 3015.49 | 300.56 | 245.21 | 1655.18 | 661.25 | 1134.70 | 2509.35 | 1048.17 |

*Table 11.* **Running time on an NVIDIA L40s GPU.** We present the average frames per second (FPS) for each task.

## D. Simulation Performance

We benchmark the running time of our simulator in Table 11. Note that the frame time in the simulation is 1e-3 seconds.

We further examine how well our simulator aligns with fundamental conservation laws, focusing specifically on the conservation of momentum during two distinct contact scenarios, as illustrated in Figure 12. The scenarios we simulate are "Parallel Contact" and "Inclined Contact." The corresponding plots on the right-hand side track the system's total momentum error over time. It is observed that the error magnitude ranges from approximately 1e-13 to 1e-14, indicating that our simulator effectively conserves momentum during these contact events.

## E. Real-World Relevance of Benchmark Tasks

In Section 4.2, we detail the DLO-Lab benchmark, designed to capture the diverse spectrum of real-world DLO manipulation challenges. Unlike prior works that often rely on abstract toy problems, we selected tasks that map directly to fundamental manipulation primitives essential in industrial manufacturing, logistics, and surgery. We explicitly connect our simulation scenarios to the following five core skills:

1. **Cable Routing & Threading**. Essential for industrial harnessing and surgical suturing, tasks like **Wiring-post** and **Wiring-ring** require guiding DLO tips through constrained apertures and specific channels with high precision.

2. **Topological Manipulation**. Critical for cable maintenance and logistics, **Unknotting** and **Separation** demand reasoning about global topology to resolve complex knots and disentangle cluttered structures.

3. **Winding & Binding**. Mirroring coil winding and packaging processes, **Coiling** and **Wrapping** necessitate precise tension management to ensure DLOs conform tightly to rigid geometries.

4. **Precision Shape Control**. Representing applications like wire bending for circuitry or catheter shaping, **Letter Art** challenges agents to deform DLOs into strict target geometries rather than loose functional configurations.

5. **DLO-Mediated Manipulation**. Simulating tool use in scenarios like harvesting or construction, **Gathering**, **Lifting**, and **Slingshot** leverage the DLO's tension and shape to actuate, constrain, or propel other objects.

Collectively, these tasks provide a rigorous and comprehensive testbed, ensuring that policies trained within our simulator possess the robust, transferable skills necessary for complex real-world deployment.

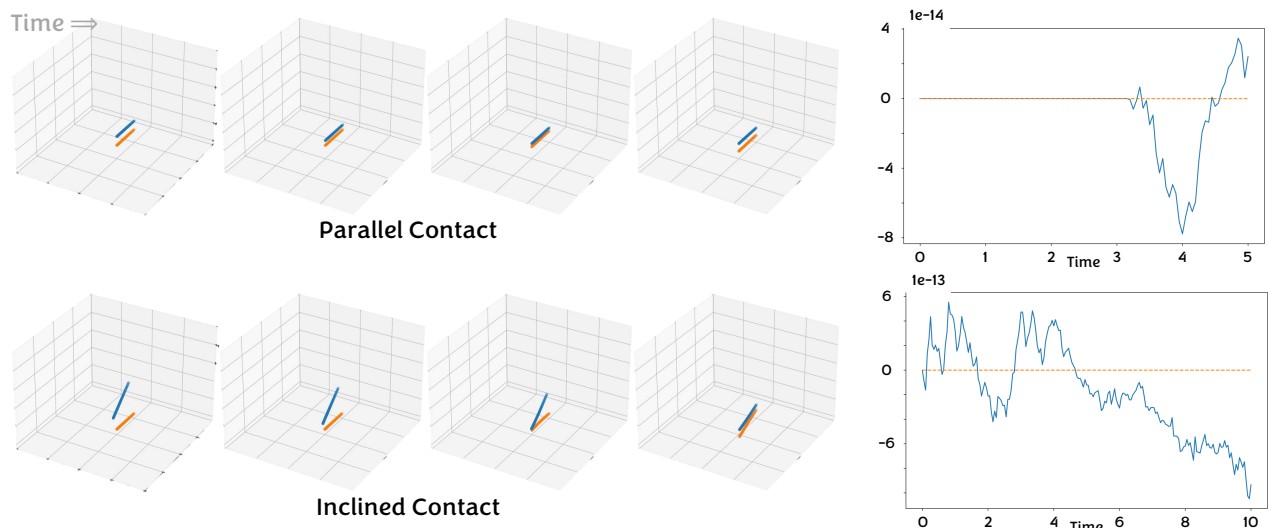

Figure 12. **Analysis of momentum conservation.** On the left, we display the contact cases we considered. On the right, we plot the system's total momentum error over time. Our simulator conserves momentum with very low errors during contact events.

## F. Detailed Prompts of the DLO Agent

Here, we provide the prompts used in our experiments for the two functionalities of the DLO agent. We use the Gemini-3-Pro-Preview model in this work.

### F.1. Grasp Proposal

**Candidate Mode** Below is the prompt we use for the *Candidate* mode for the grasp proposal.

```
You are an intelligent AI assistant for robotics, physical simulation, and deformable object manipulation.
Follow the user's requirements carefully and make sure you understand them.
Keep your answers short and to the point.
Do not provide any information that is not required.

You will suggest grasping points for solving a manipulation task involving deformable linear objects (DLO),
ensuring they are both efficient and robust for the task.
Here is the task description and reward definition:
{task_description}
{reward_description}

The dot markers with integer numbers in the attached images indicate the candidate grasping points on the DLO.
For this task, there are **{n_robots}** robots.
Here we calculate the distance from each grasping point to the basement of each robot:
{distance_info}

Using the information provided and your understanding of deformable manipulation, please recommend grasping
points for each robot involved in the task.
Please note that the grasping points will **not** be changed during the manipulation, so they should be chosen
carefully to ensure successful task completion.

Please provide your answer in the following JSON format:
```
{
    "grasping_point_for_robot_1": ... # An integer,
    "grasping_point_for_robot_2": ... # (Optional) If applicable, an integer,
    "reasoning_for_the_suggestion": ... # Provide a brief reasoning for your suggestion.
}
```

The output should **only** contain the JSON dictionary.
```

**Coefficient Mode** Below is the prompt we use for the *Coefficient* mode for the grasp proposal.

```
You are an intelligent AI assistant for robotics, physical simulation, and deformable object manipulation.
Follow the user's requirements carefully and make sure you understand them.
Keep your answers short and to the point.
```

```
Do not provide any information that is not required.

You will suggest grasping points for solving a manipulation task involving deformable linear objects (DLO),
ensuring they are both efficient and robust for the task.
Here is the task description and reward definition:
{task_description}
{reward_description}

The start and the end markers in the attached images indicate the starting vertex and the ending vertex of the
DLO, respectively.
For this task, there are **{n_robots}** robots.
Here we calculate the distance from start or end vertex to the basement of each robot:
{distance_info}

Using the information provided and your understanding of deformable manipulation, please recommend grasping
points for each robot involved in the task.
The grasping points should be reprsented as coefficients in [0, 1], where 0 corresponds to the starting vertex
and 1 corresponds to the ending vertex of the DLO.
Please note that the grasping points will **not** be changed during the manipulation, so they should be chosen
carefully to ensure successful task completion.

Please provide your answer in the following JSON format:
```
{
    "grasping_point_for_robot_1": ... # A float in [0, 1],
    "grasping_point_for_robot_2": ... # (Optional) If applicable, a float in [0, 1],
    "reasoning_for_the_suggestion": ... # Provide a brief reasoning for your suggestion.
}
```

The output should **only** contain the JSON dictionary.
```

**Marker Mode**  Below is the prompt we use for the *Marker* mode for the grasp proposal.

```
You are an intelligent AI assistant for robotics, physical simulation, and deformable object manipulation.
Follow the user's requirements carefully and make sure you understand them.
Keep your answers short and to the point.
Do not provide any information that is not required.

You will suggest grasping points for solving a manipulation task involving deformable linear objects (DLO),
ensuring they are both efficient and robust for the task.
Here is the task description and reward definition:
{task_description}
{reward_description}

The start and the end markers in the attached images indicate the starting vertex and the ending vertex of the
DLO, respectively.
For this task, there are **{n_robots}** robots.
Here we calculate the distance from start or end vertex to the basement of each robot:
{distance_info}

Using the information provided and your understanding of deformable manipulation, please recommend grasping
points for each robot involved in the task.
You need to mark the grasping points on the DLO in the **first** image by providing the image coordinates (x, y)
in pixels (using OpenCV coordinate system).
For reference, the start and end vertex in the image have the following coordinates:
{coordinates_info}
Please note that the grasping points will **not** be changed during the manipulation, so they should be chosen
carefully to ensure successful task completion.

Please provide your answer in the following JSON format:
```
{
    "grasping_point_for_robot_1": ... # A tuple (x, y) representing image coordinates in pixels of the **first**
    image,
    "grasping_point_for_robot_2": ... # (Optional) If applicable, a tuple (x, y) representing image coordinates
    in pixels of the **first** image,
    "reasoning_for_the_suggestion": ... # Provide a brief reasoning for your suggestion.
}
```

The output should **only** contain the JSON dictionary.
```

## F.2. Task Decomposition

**Initial Decomposition Plan**  Here is the prompt we use for obtaining the initial decomposition plan.

```
You are an intelligent AI assistant for robotics, physical simulation, and deformable object manipulation.
Follow the user's requirements carefully and make sure you understand them.
Keep your answers short and to the point.
Do not provide any information that is not required.

You will divide a manipulation task for deformable linear objects into the fewest possible subtasks, keeping the
robot gripper's grasping point on the linear object unchanged in each subtask.
That is, break down the task into subtasks when a switch in grasping points is required to complete it.

Here is the overall task description:
{task_description}

Here is the code snippet for the task environment setup:
{task_environment}

Using the information provided above, and the attached rendered images of the scene, decompose the task into the
fewest possible subtasks.
Your response should be a list of dictionaries in JSON format, where each dictionary contains:
- "subtask_id": An integer representing the subtask number (starting from 1).
- "subtask_description": A brief description of the subtask.
- "reward_function": A code snippet defining the reward function for the subtask using the provided environment
setup.
- "reward_description": A brief description in natural language of the reward function.
- "horizon_length": An integer specifying the maximum number of action steps allowed to complete the subtask.

Finally, you should append a dictionary with one key-value pair describing the overall task completion criteria
after the list of subtasks, i.e.,
- "overall_task_completion_criteria": A brief description in natural language of the criteria for successfully
completing the overall task.

Ensure that the subtasks are sequential and collectively achieve the overall task goal.

Make sure to format your response as valid JSON. The output should **only** contain the JSON list of subtasks
and the overall task completion criteria.
```

**Plan Update**  Here is the prompt we use for evaluation and plan update after completing the optimization of each sub-task.

```
You are an intelligent AI assistant for robotics, physical simulation, and deformable object manipulation.
Follow the user's requirements carefully and make sure you understand them.
Keep your answers short and to the point.
Do not provide any information that is not required.

You are working on subtask {subtask_id} of a larger manipulation task for deformable linear objects.
Here is the overall task description:
{task_description}

Here is the code snippet for the task environment setup:
{task_environment}

The original task decomposition plan for the whole task is as follows:
{previous_plan}

Given the current state of the environment, you should:
(1) first judge whether the overall task has been completed successfully after executing subtask {subtask_id};
(2) if the overall task is not yet completed, update subsequent subtasks in the previous plan as necessary to
ensure successful completion of the overall task.

Specifically, here is the optimized trajectory from the robot's execution of subtask {subtask_id} (attached as
rendered videos).
Here is the final state of the environment after executing subtask {subtask_id}:
{subtask_final_state}

Using the information above, you may update the subtasks following subtask {subtask_id} to account for any
changes in the environment or object state resulting from the execution of subtask {subtask_id}.
Your response should be a list of dictionaries in JSON format,
(1) if the overall task has been completed successfully, return a list with a single dictionary:
- "reasoning_for_completion": A brief explanation in natural language of why the overall task is considered
completed successfully after executing subtask {subtask_id};
(2) otherwise, return the updated plan, i.e., the list of dictionaries where each dictionary contains:
- "subtask_id": An integer representing the subtask number (starting from {subtask_id + 1}, representing the
next and subsequent subtasks).
- "subtask_description": A brief description of the subtask.
- "reward_function": A code snippet defining the reward function for the subtask using the provided environment
setup.
- "reward_description": A brief description in natural language of the reward function.
- "horizon_length": An integer specifying the maximum number of action steps allowed to complete the subtask.
Ensure that the subtasks are sequential and collectively achieve the overall task goal.

Make sure to format your response as valid JSON. The output should **only** contain the JSON list of subtasks
and the overall task completion criteria.
```

## G. Future Directions

While DLO-Lab establishes a comprehensive platform for learning deformable manipulation, our benchmark results indicate that no single optimization strategy currently suffices for all tasks. Gradient-based methods are efficient but tend to struggle with non-smooth landscapes, whereas sampling-based approaches provide robustness but come with higher computational costs. A promising direction for future research is the development of hybrid optimization algorithms that combine the strengths of these two methods. For example, integrating zeroth-order sampling with first-order analytic gradients could effectively balance the bias-variance trade-offs often encountered in differentiable physics, as suggested by (Suh et al., 2022). Additionally, we believe an interesting area for exploration is enhancing the DLO agent's capabilities to overcome the specific limitations of gradient-based trajectory optimization methods in indirect manipulation tasks. By leveraging the agent to plan initialization trajectories that create stable contact, we can address challenges in sparse-reward scenarios, such as tool use. This approach will allow the optimizer to take over once effective gradient flows are established. Finally, while we have demonstrated zero-shot sim-to-real transfer in both open-loop and closed-loop settings, improving the robustness and coverage of this pipeline remains an open challenge. In particular, more reliable real-time DLO state estimation across diverse objects and environments, as well as scaling synthetic demonstration generation for broader visuomotor policy training, are promising avenues for future work.

