# OpenReview forum: "DLO-Lab: Benchmarking Deformable Linear Object Manipulations with Differentiable Physics"
_ICML.cc/2026/Conference — ICML 2026 regular_

### Official Review · Reviewer_m4F8 · 2026-03-09

**Soundness:** 3
**Presentation:** 4
**Significance:** 3
**Originality:** 3
**Overall Recommendation:** 4
**Confidence:** 3

**Summary:**

This paper proposes DLO-Lab, a simulation platform and benchmark suite for deformable linear object (DLO) manipulation based on differentiable physics. It validates the effectiveness of the benchmark through various tasks and baseline algorithms, and ultimately conducts effective testing in real physical environments, providing a new generalized foundation for research on robotic DLO manipulation.

**Compliance With Llm Reviewing Policy:**

Affirmed.

**Ethical Review Concerns:**

Concerns regarding data generation and the application of VLA models have been addressed, and I have made adjustments to the scores.

**Final Justification:**

Concerns regarding data generation and the application of VLA models have been addressed, and I have made adjustments to the scores.

**Key Questions For Authors:**

1. Currently, several works based on two-layer planner schemes have also achieved the manipulation of deformable objects [1][2][3], and these schemes appear to exhibit stronger generalization ability. It is hoped that the authors could conduct an analysis and comparison of these schemes.
2. The System Identification section for Sim2Real in the appendix is, in my view, a key component. I suggest that the authors supplement more quantitative results in the main body of the paper to demonstrate the advantages of DLO-Lab, rather than only presenting the performance of existing strategies. For instance, quantitative data for different materials in the same task.
3. Could the authors supplement the quantitative results of closed-loop policies implemented in real-world scenarios?

Depending on the author’s response, I may revise my rating upwards.

[1]Learning Generalizable Language-Conditioned Cloth Manipulation from Long Demonstrations

[2]DexGarmentLab: Dexterous Garment Manipulation Environment with Generalizable Policy

[3]GR-RL: A Breakthrough in Dexterous Manipulation, Achieving the First Real-Robot Shoe Lacing with Reinforcement Learning

**Limitations:**

It is suggested that the authors supplement the discussion on the limitations of the paper, including that the DLO agent relies on candidate position design and thus lacks generality, and that the Sim2real section fails to address the performance of closed-loop policies.

**Strengths And Weaknesses:**

Strengths:
1. DLO-Lab has established a highly instructive framework in the field of deformable object simulation, boasting sufficient visual fidelity and physical realism.
2. This paper is well-structured with coherent and fluent expression, rendering it easy to understand.

Weaknesses:
1. The necessity of the VLM. The agent design in this paper requires a great deal of prior knowledge and relies on structured environments, which greatly compromises the generality of DLO-Lab.
2. I have some concerns regarding the focus of this paper. While the paper purports to center on the entire DLO-Lab framework, most of the quantitative data in the experimental section is derived from the evaluation of existing reinforcement learning and optimization strategies.
3. In the Sim2Real section, the authors highlight the consistency between the simulation scheme and real-world physical phenomena; yet, the evaluation of policies is not validated using the closed-loop policies native to DLO-Lab.
4. In terms of baseline selection, the authors adopt two-layer planning schemes based on prior knowledge. However, several existing works employing such two-layer planning schemes appear to exhibit stronger generalization ability, which diminishes the advantages of the VLM+RL and optimization strategies in DLO-Lab.

---

> ### Author Rebuttal · Authors · 2026-03-31
>
> Thank you for the valuable comments. We try to address your concerns below.
>
> For the `visual results` mentioned, please see this [anonymous link](https://anonymous.4open.science/r/DLO-Lab-rebuttal-figures-6301).
>
> > W1: The necessity of the VLM.
>
> Our VLM agent is not intended as a final closed-loop policy but rather acts as an automated "expert demonstrator" to overcome challenges in simulation exploration. By injecting structural priors (e.g., grasp points, task decomposition) to guide the optimizer, the VLM enables DLO-Lab to act as a high-throughput data engine. Combining this diverse synthetic data with real-world datasets provides the large-scale corpus necessary to train truly generalizable Vision-Language-Action (VLA) models.
>
> > W2: Focus of this paper.
>
> We would like to clarify that our core focus is twofold: (1) introducing a novel, differentiable multi-material simulator, and (2) establishing a comprehensive benchmark suite. We extensively evaluated existing RL and optimization baselines strictly to validate this benchmark—proving tasks are solvable, establishing baselines, and exposing algorithmic bottlenecks.
>
> Ultimately, DLO-Lab is designed as a high-fidelity data engine. By accurately modeling complex physics like bending plasticity and loop holonomy, it enables the efficient generation of massive synthetic datasets. Combined with real-world data, this corpus is critical for eventually training generalizable VLAs or WAMs capable of versatile, unstructured DLO manipulation in the real world.
>
> > W3: Closed-loop policies native to DLO-Lab.
>
> Please see `Q3` below.
>
> > W4: Comparison with existing planning baselines.
>
> Please see `Q1` below.
>
> > Q1: Analysis and comparison of [1][2][3].
>
> We thank the reviewer for highlighting these impressive works. While we agree these hierarchical schemes exhibit exceptional generalization, they do not diminish our VLM baseline; rather, they highlight the exact synergy DLO-Lab is designed to support. Below, we compare these methods with our work:
> - Differences in Generalization: Methods like [1] and [2] achieve generalization by exploiting the inherent semantic keypoints of garments (e.g., collars, sleeves). Because featureless DLOs (ropes/cables) lack these markers, our VLM compensates by injecting dynamic spatial and temporal priors based on real-time topology.
> - Baseline vs. Specialized Policy: Our VLM is not a finalized, real-world policy. It is an "expert demonstrator" designed to break exploration bottlenecks and generate the synthetic data needed to eventually train the advanced policies the reviewer mentions.
> - Discussion on GR-RL [3]: GR-RL perfectly highlights the need for DLO-Lab. To achieve shoelacing, [3] relies on expensive real-world RL because standard simulators fail to model the complex contact dynamics of a shoelace inside an eyelet. DLO-Lab provides the exact multi-material coupling required to bring such pipelines into simulation, drastically reducing real-world costs.
>
> We view these advanced schemes not as competitors, but as precisely the algorithms DLO-Lab was built to evaluate and accelerate. We will expand our Related Works section to discuss this convergence.
>
> > Q2: More results on system ID.
>
> Thanks for this constructive advice! We conducted new system ID experiments on two distinct ropes: a stiff white nylon and a flexible hemp. The table below details our quantitative results.
>
> | |Black Rope (in Figure 9)|White Rope|Hemp Rope|
> | --- | --- | --- | --- |
> |Stretching (x$10^5$)|4.86|7.12|4.02|
> |Bending (x$10^5$)|6.64|37.79|19.63|
>
> We also show visual results in `Figures E-F` in the above link. The results prove our simulator's ability to capture unique material properties, and we will integrate this expanded section into the revised main text.
>
> > Q3: Results for closed-loop policies.
>
> To evaluate real-world manipulation, we built a "project-and-unproject" pipeline to bridge the sim-to-real perception gap:
> - Real-World Observation: We HSV-threshold an RGB depth image, unprojecting mixed rope and target objects 2D pixels into an unordered 3D point cloud of size $N$.
> - Simulation Replication: We project clean 3D simulation points to 2D using known camera parameters, inject calibrated noise, unproject to 3D, and sample $N$ points.
> - Structured Noise Modeling: We apply symmetric Gaussian depth noise (sensor uncertainty) and asymmetric half-normal viewing-axis noise (matching our oblique camera angle).
> - Policy Input: The mixed point cloud is sorted along the world-frame y-axis and normalized online via Welford's algorithm for the MLP.
>
> Using this pipeline, we deployed a closed-loop PPO policy for Wiring-ring (see `Figures A-B`). The agent achieved a success rate of 58.3% across 12 trials with randomly initialized rope positions, averaging 69 action steps to complete the task, validating our pipeline's robustness.
>
> > Limitations
>
> We thank the reviewer for the constructive suggestions and will explicitly discuss them in the revised paper.

---

> > ### Author Rebuttal · Reviewer_m4F8 · 2026-04-03
> >
> > Regarding the response to W1, I believe the original paper does not include any section on generating data for training purposes. Could the authors provide evidence demonstrating whether the data provided by DLO-Lab can “truly train generalizable Vision-Language-Action (VLA) models”，for instance, the actual performance of VLA policies? Providing such evidence would help address some limitations of the current manuscript.
> >
> > In the response to W2, the authors outlined two core focuses. First, could the authors clarify whether the “differentiable multi-material simulator” introduces any innovations beyond engineering improvements compared to the physical engine framework of the original simulator? Additionally, concerning the benchmark suite, all methods evaluated in the current paper are existing ones, and integrating them appears to be another engineering-oriented effort.
> >
> > I agree with the authors’ responses to Q2 and Q3. Overall, the current paper reads largely as an engineering integration of existing modules. I recommend that the authors further supplement validation related to the dataset and end-to-end policy models such as VLAs. This would help the paper overcome its current limitations.

---

> > > ### Author Response · Authors · 2026-04-07
> > >
> > > We sincerely appreciate the reviewer's continued dialogue and affirmation of our responses to Q2 and Q3. Below, we address your remaining points concerning the scope and contributions of our work:
> > >
> > > 1. **Data Generation and VLA Training**
> > >
> > > We sincerely thank the reviewer for their constructive suggestion. We agree that including empirical validation of VLA significantly enhances the manuscript. During the rebuttal period, we successfully conducted this experiment to demonstrate DLO-Lab's capability as a high-quality data engine:
> > >
> > > - **Experimental Setup**: Using the trained policies in our DLO-Lab, we generated 800 successful demonstrations across 4 manipulation tasks (200 demos per task). Each episode records rich multi-view visual observations (a front camera and 1-2 wrist cameras depending on the task, see `Figure G` in the [anonymous link](https://anonymous.4open.science/r/DLO-Lab-rebuttal-figures-6301) for some examples), along with robot proprioceptive states and delta actions. We then finetuned the action expert of SmolVLA, an efficient Vision-Language-Action model, directly on these combined demonstrations for 40K iterations with a batch size of 128.
> > >
> > > - **Results**: We report the final task success rates below. We also visualize successful VLA inference in `Figure H` in the [anonymous link](https://anonymous.4open.science/r/DLO-Lab-rebuttal-figures-6301). Please note that the baseline RL methods require a separate, dedicated policy trained from scratch for each task, whereas **SmolVLA is evaluated as a single multi-task policy**.
> > >
> > > ||Coiling|Separation|Unknotting|Wiring-post|Average|
> > > |:---:|:---:|:---:|:---:|:---:|:---:|
> > > |PPO|67%|100%|0%|67%|58.5%|
> > > |SAC|0%|100%|0%|0%|25.0%|
> > > |SHAC|93%|100%|73%|0%|66.5%|
> > > |SAPO|100%|100%|80%|0%|70.0%|
> > > |SmolVLA[a]|60%|100%|73%|7%|60.0%|
> > >
> > > [a] Shukor, Mustafa, et al. Smolvla: A Vision-Language-Action Model for Affordable and Efficient Robotics. arXiv:2506.01844 (2025).
> > >
> > > These results strongly confirm that DLO-Lab produces data suitable for training generalizable visual policies. Additionally, they emphasize the high informational density and quality of the data rendered by our engine.
> > >
> > > We will include the entire VLA training section and the accompanying analysis in the revised manuscript to explicitly highlight the benefits of using DLO-Lab for generating high-fidelity synthetic data.
> > >
> > >
> > > 2. **Contribution Beyond Engineering Improvement**
> > >
> > > We appreciate the reviewer acknowledging the heavy system-building aspect of our work. However, we respectfully argue that bridging the gap between isolated physics theories and a unified, differentiable RL environment requires overcoming severe mathematical and algorithmic bottlenecks. We highlight our core methodological innovations below:
> > >
> > > - **Advancing DLO Solver Theory (Mathematical Innovation)**: Standard Discrete Elastic Rod (DER) theory is insufficient for realistic manipulation, as it only covers stretching, bending, and twisting behaviors. To build a high-fidelity, differentiable solver, we had to fundamentally extend this theory. As detailed in Section A.1 and Eq. 10-13, we mathematically formulated and integrated (1) complex self-collision handling, (2) inextensibility constraints, and (3) bending plasticity in our customized solver. Furthermore, coupling discrete DLOs with continuum-based soft materials requires a fundamentally new algorithm (Section 3.2), not mere software integration. We designed a novel Eulerian grid-mediated coupling scheme within the Material Point Method (MPM). By calculating symmetric repulsive impulses and applying opposite reaction forces via atomic operations, our formulation guarantees strict momentum conservation and stable, immediate bidirectional feedback within a single simulation step.
> > >
> > > - **The DLO Agent (Algorithmic Innovation)**: End-to-end RL and trajectory optimization fail on long-horizon topological tasks due to massive exploration spaces. We designed a novel DLO Agent (Section 4.3) that introduces dynamic spatial priors (grasp proposals) to filter continuous action spaces, and temporal priors (task decomposition) to handle sequential dependencies. This is a fundamentally new algorithmic approach to topology-constrained manipulation.
> > >
> > > - **"Project-and-Unproject" Sim-to-Real Pipeline (Perception Innovation)**: Rather than relying on perfect state information, we developed a novel structured noise-injection pipeline that mathematically models the exact failure modes of depth cameras (symmetric uncertainty and asymmetric mounting spread). This methodological innovation allows policies trained purely in DLO-Lab on unsegmented point clouds to achieve zero-shot, closed-loop deployment in the real world.
> > >
> > > We believe that extending the original DER theory to encompass broader DLO dynamics, designing novel topology-aware planners, and standardizing evaluation metrics represent significant scientific contributions beyond pure system integration.

---

### Official Review · Reviewer_PFLK · 2026-03-09

**Soundness:** 3
**Presentation:** 3
**Significance:** 2
**Originality:** 2
**Overall Recommendation:** 3
**Confidence:** 4

**Summary:**

The paper introduces DLO-Lab, a differentiable simulation and benchmark suite that is built on Taichi and Genesis and designed for deformable linear object (DLO) manipulation. DLO-Lab integrates DER and MPM solvers to model complex DLO dynamics, including bending plasticity, loop topologies, and multi-material coupling. The authors propose a benchmark of 10 manipulation tasks. To address the exploration challenges of long-horizon tasks, they introduce a VLM-based agent for proposing grasping points and dynamically decomposing tasks. The empirical study evaluates model-free RL, first-order model-based RL, and trajectory optimization methods. It is suggested that sample-based trajectory optimization (CMA-ES) currently achieves the highest reliability and validates policies via zero-shot sim-to-real transfer.

**Compliance With Llm Reviewing Policy:**

Affirmed.

**Final Justification:**

My main concern remains that the paper’s central contribution is still not clearly distinguished from "adding DLO support to Genesis". The discussion helped clarify some important points, especially the Eq. 7 modeling issue, the robustness of the VLM grasp proposal, and several possible uses of differentiability, but these mostly strengthen the implementation story rather than fully establish a broader methodological contribution. What still remains unresolved is convincing evidence that DLO-Lab delivers a qualitatively new capability or insight beyond integrating DLO simulation into an existing differentiable engine stack, and that this added differentiability leads to a practically compelling advantage over strong existing simulators for the core manipulation problems studied.

**Key Questions For Authors:**

- Given that the VLM agent relies on 2D rendered images to propose 3D grasping points, how sensitive is the Candidate mode to the chosen camera angle or partial occlusions?
- What is the core practical advantage of using DLO-Lab over a highly optimized, non-differentiable simulator, or other existing differentiable engines? If the gradients are too rugged to use for complex policy learning, could you provide comparative evidence demonstrating that DLO-Lab offers either superior computational speed (wall-clock time) or significantly better physical fidelity (e.g., a tighter sim-to-real gap) than established alternatives?

**Limitations:**

yes

**Strengths And Weaknesses:**

**Strengths**

- The comprehensive framework integrates DER and MPM solvers to achieve stable two-way coupling between co-dimensional DLOs and soft/rigid bodies. It goes beyond basic elasticity to natively support critical real-world phenomena like bending plasticity and closed-loop holonomy.
- The benchmark suite features 10 distinct tasks mapping directly to real-world applications such as cable routing, tool use, and knot untangling. The inclusion of zero-shot sim-to-real hardware experiments demonstrates that the simulator captures sufficient physical fidelity to bridge the reality gap.
- The introduction of a VLM-driven agent to propose task-specific grasp points mitigates the combinatorial explosion of the action space.

**Weaknesses**

- The paper's primary empirical finding, that sample-based trajectory optimization (CMA-ES) outperforms gradient-based methods due to discontinuous contact landscapes, is a well-documented phenomenon in differentiable physics (e.g., [1-5]). If the best-performing baseline (CMA-ES in this case) bypasses analytical gradients entirely because the optimization landscape is "notoriously rugged", the fundamental motivation for building a differentiable simulator is undermined.
- The paper also fails to clearly establish the practical value proposition of DLO-Lab over existing physics engines. If gradients are ultimately discarded or unusable for complex, tool-use tasks, it remains unclear why a researcher would choose this simulator over highly optimized, non-differentiable engines (e.g., MuJoCo, Isaac Sim), or even over other off-the-shelf differentiable simulators (like those listed in Table 1) whose gradient limitations are already known. Proving that DLO-Lab can compute gradients for complex topologies and coupling is only impactful if those gradients actually improve downstream manipulation.
- In Equation 7, the bending stiffness matrix $B_i$ is presented as a diagonal matrix parameterized purely by radius $r_i$. It is not mathematically justified how this isotropic assumption generalizes when heterogeneous or non-circular DLOs (mentioned as a feature) are used.

**References**

[1] Metz et al., 2021 (https://arxiv.org/abs/2111.05803) \
[2] Suh et al., ICML 2022 (https://proceedings.mlr.press/v162/suh22b/suh22b.pdf) \
[3] SHAC, ICLR 2022 (https://short-horizon-actor-critic.github.io/) \
[4] Turpin et al., ICRA 2023 (https://dexgrasp.github.io/) \
[5] SAPO, ICLR 2025 (https://rewarped.github.io/)

---

> ### Author Rebuttal · Authors · 2026-03-31
>
> We thank the reviewer for their instrumental comments. Here are our responses to the concerns.
>
> For the `Figure` or `Table` mentioned, please see this [anonymous link](https://anonymous.4open.science/r/DLO-Lab-rebuttal-figures-6301).
>
> > W1: Motivation for a differentiable simulator.
>
> The motivation for DLO-Lab extends well beyond pure gradient-based trajectory optimization. Our differentiable simulator provides four fundamental advantages over non-differentiable engines (e.g., MuJoCo):
>
> - System ID: Gradients enable highly efficient sim-to-real calibration by backpropagating real-world visual projection errors directly to physical parameters.
> - Efficiency in Smooth Landscapes: When tasks lack severe discontinuities, gradient descent (GD) is vastly faster. For example, GD solves Coiling in ~8 minutes (vs. 30 minutes for CMA-ES) and outperforms 6.5 hours of CMA-ES sampling on Separation in just 8 minutes (Figure 7).
> - Enabling Closed-Loop FO-MBRL: Unlike open-loop CMA-ES, analytic gradients empower First-Order Model-Based RL to learn closed-loop policies, solving precision-critical tasks (Coiling, Unknotting) where standard sample-based RL fails.
> - Hybrid Trajectory Optimization: We developed a hybrid method combining CMA-ES exploration with GD exploitation. By using GD to refine the top 10% of CMA-ES trajectories, we successfully enhance final returns and reduce variance across benchmarks (`Table A`), proving the value of gradients even in contact-rich environments.
>
> > W2: Practical value proposition.
>
> We clarify that gradients in DLO-Lab are not discarded; they actively improve downstream manipulation via automated system ID, FO-MBRL, and hybrid trajectory optimization. Furthermore, DLO-Lab overcomes critical limitations in existing physics engines:
>
> - Coupling: Standard engines (MuJoCo) lack native DLO dynamics, while differentiable simulators (DaXBench) lack the two-way multi-material coupling required for manipulation. Simulators that do support coupling (C-IPC) require prohibitive simulation times. DLO-Lab uniquely overcomes all these bottlenecks (see `Figure C` for visual results).
>
> - Speedup: Unlike CPU-bound simulators (PyElastica) that bottleneck RL workflows, DLO-Lab’s massive GPU parallelization achieves a 220x speedup at 500 environments (`Table B`), making sample-intensive policy learning highly tractable.
>
> > W3: Equation 7.
>
> We thank the reviewer for this observation. While Eq. 7 assumes a circular cross-section and isotropic bending, we will update the manuscript to clarify how our framework supports both:
> (1) Anisotropic DLOs: To model flat cables, we generalize radius $r_i$ to an ellipse with semi-axes $a_i$ and $b_i$, natively capturing anisotropic area moments of inertia.
> (2) Heterogeneous Materials: For compositional variation along the DLO's length, our per-segment solver (Section. A.1) seamlessly assigns localized parameters (e.g., stiffness, mass) to specific sections.
>
> > Q1: Sensitivity of grasp proposal.
>
> Our Candidate mode mitigates viewpoint sensitivity by grounding 2D visual inputs with invariant 3D geometric data. Specifically, we inject the 3D distance from each candidate point to the robot's base directly into the text prompt ({distance_info}, Section F.1). This provides a stable spatial anchor that is highly robust to camera shifts and occlusions.
>
> To empirically validate this, we tested the Candidate mode on 10 pairs of randomly selected camera views across three tasks. As shown in `Figure D`, the VLM's grasp proposals (colored circles) remain highly concentrated and consistent despite viewpoint variations.
>
> > Q2: Practical advantage.
>
> DLO-Lab’s core practical advantage lies in its unique combination of differentiability, comprehensive physical coupling, and high-throughput parallelization—a triad currently unsupported by existing platforms:
>
> - High-Fidelity DLO Dynamics: Standard rigid-body engines lack native support for deformable linear objects. DLO-Lab provides a high-fidelity DLO simulation that accurately captures complex physical phenomena.
> - Comprehensive Soft/Rigid Coupling: Unlike many differentiable engines limited to isolated soft bodies, DLO-Lab natively supports stable, two-way coupling between DLOs, rigid bodies (e.g., grippers), and other soft bodies—a strict prerequisite for realistic manipulation. (`Figure C`)
> - System ID: Non-differentiable engines (e.g., MuJoCo) require inefficient manual or black-box tuning. DLO-Lab leverages analytical gradients to directly optimize physical parameters (stiffness, friction, mass) against real-world data, yielding digital twins with a significantly tighter sim-to-real gap. (Figure 9 and `Figures E-F`)
> - GPU Parallelization: DLO-Lab’s massive GPU parallelization drastically reduces wall-clock time for sample-intensive RL, achieving a ~200-fold FPS speedup over PyElastica when scaling to 500 parallel environments. (`Table B`)
>
> We will explicitly highlight these comparative advantages in the revised manuscript.

---

> > ### Author Rebuttal · Reviewer_PFLK · 2026-04-03
> >
> > Thank the authors for the detailed rebuttal. The rebuttal addresses several of my concerns. I appreciate the clarification that differentiability is useful not only for direct gradient-based trajectory optimization but also for other tasks. The additional camera-view robustness analysis is helpful, and the clarification of Eq. 7 resolves my technical concern there.
> >
> > However, some concerns remain. Table A shows that the hybrid method still underperforms pure CMA-ES on several tasks, and the confidence intervals largely overlap where it nominally improves. The throughput comparison against CPU-only PyElastica conflates the gains of GPU parallelization with the contributions of DLO-Lab itself. As a result, I still do not see a compelling reason why a robotics researcher would choose DLO-Lab over existing, well-optimized alternatives.

---

> > > ### Author Response · Authors · 2026-04-07
> > >
> > > We sincerely thank the reviewer for the continued engagement and for acknowledging our clarifications regarding differentiability, camera robustness, and Eq.7 in the manuscript. We address your remaining concerns below:
> > >
> > > 1. **Performance of Hybrid Method (Table A)**
> > >
> > > We do not claim that the hybrid method universally surpasses pure CMA-ES; rather, it showcases a unique capability unlocked by DLO-Lab. By providing analytical gradients, it significantly enhances performance in tasks that benefit from local gradient exploitation—such as Separation, where a smoother contact landscape allows gradients to greatly improve optimization (`Table A` in the [anonymous link](https://anonymous.4open.science/r/DLO-Lab-rebuttal-figures-6301)).
> > >
> > > Moreover, the advantage of these gradients extends beyond open-loop trajectory optimization into **closed-loop policy learning**. DLO-Lab's differentiability facilitates the training of first-order model-based RL policies, such as SHAC and SAPO. This results in substantial performance improvements (Table 8). For example, in the Unknotting task, SAPO achieves over 80% success, while standard zeroth-order RL fails entirely. In the Coiling task, SAPO achieves 100% success, whereas PPO saturates at 67% and SAC fails.
> > >
> > > This leap in capability is only possible due to DLO-Lab’s differentiable architecture. As discussed in Section G, DLO-Lab provides the **essential foundation** for future research on integrating zeroth-order sampling with first-order gradients, ultimately enhancing both the reliability and efficiency of closed-loop manipulation policies.
> > >
> > > 2. **A Compelling Reason for Choosing DLO-Lab**
> > >
> > > The compelling reason to choose DLO-Lab is that it fills an empty intersection in current simulation platforms, resolving a strict trade-off researchers currently face:
> > >
> > > - Mainstream engines (e.g., MuJoCo, Isaac Sim, Genesis): These prioritize RL speed but lack native DLO dynamics. They force researchers to approximate cables as articulated chains of rigid capsules, completely **losing critical physical properties** like stretching, twisting, and bending plasticity.
> > >
> > > - High-fidelity DLO solvers (e.g., PyElastica): These prioritize accuracy but are restricted to serial CPU execution, **making large-scale RL intractable**. Furthermore, they lack the analytical gradients required for first-order policy training and do not support two-way coupling between DLOs and other soft materials.
> > >
> > > As demonstrated in Table 1, DLO-Lab is currently the only engine that unifies the physical fidelity of DLO solvers, stable multi-material coupling, analytical gradients for first-order RL, and the massive GPU throughput required for large-scale RL training and synthetic data generation.
> > >
> > > While we believe this covers the dominant paradigms, we would be more than happy to explicitly compare DLO-Lab against any other specific "well-optimized alternatives" the reviewer has in mind in our final revision.

---

### Official Review · Reviewer_a3Fd · 2026-03-17

**Soundness:** 3
**Presentation:** 3
**Significance:** 3
**Originality:** 3
**Overall Recommendation:** 4
**Confidence:** 3

**Summary:**

In this paper, the authors aim to address a central concept in robotic manipulation: learning and benchmarking manipulation of deformable linear objects (DLOs) such as ropes and cables. The paper introduces DLO-Lab, a differentiable simulation environment that models diverse DLO material properties (e.g., elasticity, plasticity, topology) and supports coupling with rigid and soft objects. On top of this, it provides a benchmark suite of manipulation tasks and a DLO agent for grasp selection and task decomposition. The authors evaluate multiple learning paradigms (RL, model-based RL, trajectory optimization), showing that sampling-based optimization is most robust, while gradient-based methods are efficient but brittle. They also demonstrate sim-to-real transfer, suggesting the simulator has sufficient physical fidelity. Overall, the authors analyze a notable theme: trade-offs between differentiability, realism, and optimization in deformable manipulation.

**Compliance With Llm Reviewing Policy:**

Affirmed.

**Key Questions For Authors:**

- Do policies trained in DLO-Lab generalize across tasks, materials, or initial conditions?
- How robust is the approach under partial or noisy observations instead of full simulator state?
- How would results change if trajectory optimization methods were required to produce closed-loop policies?
- What is the individual contribution of grasp proposal vs task decomposition?

**Limitations:**

yes

**Strengths And Weaknesses:**

### Strengths
- The paper is well written and organized.
- Tasks presented in this paper are diverse, realistic, and capture both geometric and topological challenges, going beyond toy problems.
- The proposed simulator combines differentiability, material diversity, and multi-material coupling, which prior work lacks.

### Weaknesses
- The core contributions are largely system integration rather than fundamentally new methods.
- The method relies on full observations, it is not clear how the method generalizes across tasks or real-world settings.
- The comparison between trajectory optimization and policy learning is not entirely fair, as trajectory optimization solves an open-loop control problem, whereas policy learning requires closed-loop generalization and robustness.

---

> ### Author Rebuttal · Authors · 2026-03-31
>
> Thanks for identifying our work and providing valuable comments. We try to address your concerns below.
>
> For the `Figure` mentioned, please see this [anonymous link](https://anonymous.4open.science/r/DLO-Lab-rebuttal-figures-6301).
>
> >  W1: Contribution.
>
> While DLO-Lab involves significant integration, unifying isolated physics solvers into parallelized RL pipelines resolves severe architectural bottlenecks. We respectfully argue this is a scientific contribution, alongside several novel components:
>
> - Overcoming Bottlenecks: Just as standardizing rigid-body physics (e.g., MuJoCo) founded locomotion RL, our unified, differentiable, GPU-parallelized engine for multi-material coupling is a critical prerequisite for DLO manipulation research.
> - New Planning Method: Our novel DLO agent (Section 4.3) injects dynamic spatial (grasp proposals) and temporal (task decomposition) priors to guide the optimizer, solving intractable long-horizon local minima.
> - Closed-Loop Deployment: We developed a structured noise-injection pipeline modeling depth camera failures, enabling policies trained on unsegmented point clouds to achieve closed-loop real-world deployment (`Figures A-B`).
> - Benchmarking: DLO-Lab provides a testbed to fairly compare trajectory optimization and closed-loop (FO-MB)RL in this domain.
>
> > W2: Full observation was used.
>
> We would like to clarify how our framework addresses both the theoretical benchmark and practical real-world settings:
>
> - Full Observations as a Benchmark Baseline: In the simulation benchmark, we provide full state observations to the baseline algorithms deliberately to isolate the manipulation and control bottleneck from the perception bottleneck. By establishing an upper bound on performance with perfect states, we provide a clear baseline for future researchers to evaluate their own perception-in-the-loop algorithms.
>
> - Generalizing to Real-World Settings: To prove that policies trained in DLO-Lab can generalize to real-world settings without full observations, we have added a new closed-loop Sim-to-Real evaluation (`Figures A-B` in the above link). For this deployment, we explicitly drop the full-state assumption. Instead, the policy is trained and deployed using raw, noisy point clouds from a RealSense D455 depth camera with the HSV color threshold technique.
>
> > W3: Comparison fairness.
>
> We agree this is an asymmetric comparison. Our intent was not to pit them against each other, but to evaluate two complementary paradigms: (1) trajectory optimization: proves tasks are physically solvable (an "expert oracle"); (2) policy learning: distills behaviors into fast, noise-robust, reactive controllers. We will update the manuscript to emphasize that these are distinct, complementary ways to solve the benchmark rather than directly competing baselines.
>
> > Q1: Generalization.
>
>  To answer clearly, we distinguish between two types of generalization:
> - Within-Task (Materials & Initial Conditions): Yes. As demonstrated in our supplementary video, our policies are highly robust. They successfully generalize across varying physical properties (e.g., rope radius) and different starting positions without overfitting.
> - Cross-Task: Zero-shot generalization across fundamentally different tasks is out of scope for these specific baselines, which primarily serve to prove task solvability and generate expert trajectories.
>
> > Q2: Partial/noisy observation.
>
> To evaluate this, we deployed a closed-loop PPO policy on the real-world Wiring-ring task using a "project-and-unproject" sim-to-real pipeline. Instead of full states, the policy trains on noisy, partial observations simulating real depth cameras. We project clean 3D points to 2D, inject symmetric Gaussian (sensor uncertainty) and asymmetric half-normal noise (camera angle), and unproject into an unsegmented 3D point cloud. This mixed cloud (rope + target) is y-axis sorted and fed to the MLP. Trained strictly on these partial, noisy observations, the policy successfully generalized zero-shot, achieving a 58.3% real-world success rate across 12 trials (`Figures A-B`).
>
> > Q3: Closed-loop policies.
>
> Converting open-loop trajectory optimization (TO) into closed-loop policies requires implementing Model Predictive Control (MPC). This would alter results in two key ways: decreased inference speed and horizon limits.
>
> > Q4: Clarify contributions.
>
> These two components address different manipulation bottlenecks. (1) Grasp proposal (spatial prior): restricts the infinite possible grasp locations to a few strategic points, avoiding poor initial grasps that make the topology physically unrecoverable. This is vital for topology-constrained tasks (Section C.3, Table 9). (2) Task decomposition (temporal prior): breaks long-horizon, multi-grasp tasks into manageable phases with dense rewards. Dynamically updating these plans explicitly corrects execution drift, proving strictly necessary for solving sequential tasks like Wiring-ring (Section C.4, Figure 8).

---

> > ### Author Rebuttal · Reviewer_a3Fd · 2026-04-03
> >
> > Thanks for the reply. My concerns are addressed. I will keep my score unchanged.

---

> > > ### Author Response · Authors · 2026-04-07
> > >
> > > We sincerely thank the reviewer for confirming that their concerns have been addressed.
> > >
> > > In your initial review, you raised an important question (Q1) regarding whether policies trained in DLO-Lab could generalize across different tasks. While the baseline RL methods require task-specific policies, we are excited to share a new experiment conducted during the rebuttal period that directly demonstrates cross-task generalization.
> > >
> > > To prove DLO-Lab can serve as a robust data engine for generalizable policies, we collected 800 successful demonstrations across 4 distinct manipulation tasks (200 demos per task, see `Figure G` in the [anonymous link](https://anonymous.4open.science/r/DLO-Lab-rebuttal-figures-6301) for some examples). We then fine-tuned the action expert of an efficient Vision-Language-Action model (SmolVLA[a]) directly on this combined dataset for 40K iterations with a batch size of 128.
> > >
> > > [a] Shukor, Mustafa, et al. Smolvla: A Vision-Language-Action Model for Affordable and Efficient Robotics. arXiv:2506.01844 (2025).
> > >
> > > We evaluate 15 trajectories for the fine-tuned SmolVLA policy and show the success rate in the table below. As shown in the table, the single policy demonstrates cross-task generalization, achieving satisfactory performance across three out of four tasks. We also include a visualization of the VLA policy inference in `Figure H` in the [anonymous link](https://anonymous.4open.science/r/DLO-Lab-rebuttal-figures-6301).
> > >
> > > ||Coiling|Separation|Unknotting|Wiring-post|Average|
> > > |:---:|:---:|:---:|:---:|:---:|:---:|
> > > |SmolVLA|60%|100%|73%|7%|60.0%|
> > >
> > > We believe this additional result reinforces the value of DLO-Lab as a foundation for research on DLO manipulation. We will explicitly include both the initial response and the above discussion in the final revision.

---

### Decision · Program_Chairs · 2026-04-30

**Decision:**

Accept (regular)

**Comment:**

All three reviews agree that the paper presents a strong and useful systems contribution. The paper is well-written, and the proposed  DLO-Lab platform for deformable linear object manipulation is technically solid, with realistic task design, rich physical modeling, and promising sim-to-real validation. The reviewers especially value the breadth of the benchmark, the support for complex DLO (Deformable Linear Object) behaviors and multi-material interactions, and the fact that the work could become important infrastructure for future research. Overall, the paper is technically solid and relevant, with clear potential to advance the study of DLO manipulation.

The main hesitation from Reviewer PFLK is about novelty and justification. The reviewer questions whether the contribution is primarily an engineering integration rather than a fundamentally new methodological advance, and whether the differentiable nature of the simulator yields a compelling practical advantage over existing alternatives. The rebuttal helped considerably by clarifying the roles of differentiability, adding evidence on closed-loop sim-to-real transfer, system identification, and VLA training from synthetic data, and explaining the purpose of the VLM-based agent and benchmark baselines. Two reviewers appeared satisfied or partially satisfied after these clarifications, but Reviewer PFLK  remained skeptical that the paper fully proves a clear advantage over other simulators or goes beyond integration.

After reading the reviews and the discussion, it seems like this slight disagreement would be difficult to address. However, the proposed DLO-Lab is clearly a good contribution to the field of differentiable simulation for robotic manipulation.